# Differentially Private No-regret Exploration in Adversarial Markov Decision Processes

**Shaojie Bai**[1,3]     **Lanting Zeng**[1]     **Chengcheng Zhao**[1]     **Xiaoming Duan**[2]     **Mohammad Sadegh Talebi**[3]

**Peng Cheng** [*1]                                    **Jiming Chen**[1]

[1]Zhejiang University, Hangzhou, China
[2]Shanghai Jiaotong University, Shanghai, China
[3]University of Copenhagen, Copenhagen, Denmark

## Abstract

We study learning adversarial Markov decision process (MDP) in the episodic setting under the constraint of differential privacy (DP). This is motivated by the widespread applications of reinforcement learning (RL) in non-stationary and even adversarial scenarios, where protecting users' sensitive information is vital. We first propose two efficient frameworks for adversarial MDPs, spanning full-information and bandit settings. Within each framework, we consider both Joint DP (JDP), where a central agent is trusted to protect the sensitive data, and Local DP (LDP), where the information is protected directly on the user side. Then, we design novel privacy mechanisms to privatize the stochastic transition and adversarial losses. By instantiating such privacy mechanisms to satisfy JDP and LDP requirements, we obtain near-optimal regret guarantees for both frameworks. To our knowledge, these are the first algorithms to tackle the challenge of private learning in adversarial MDPs.

## 1 INTRODUCTION

Reinforcement learning (RL) is a prominent sequential decision-making framework, where an agent learns to minimize its long-term loss by interacting with an environment[1]. It has gained remarkable attraction in real-world applications across several fields such as healthcare [Gottesman et al., 2019], online recommendation [Afsar et al., 2022], and language model [Ouyang et al., 2022]. However, in these applications, the learning agent continuously improves

its performance by learning from users' personal data and feedback, which usually contain sensitive information. Without privacy protection mechanisms in place, the learning agent can memorize information of users' interaction history [Carlini et al., 2019], which makes the learning agent vulnerable to various privacy attacks [Lei et al., 2023].

Over the past decade, *differential privacy* (DP) [Dwork et al., 2006] has been extensively applied in various private decision-making settings, e.g., private multi-armed bandits [Basu et al., 2019, Tao et al., 2022]. Under DP, the learning agent collects users' raw data to train algorithms while ensuring that the output is indistinguishable from its output returned by an alternative universe where any individual user is replaced, thereby mitigating the aforementioned privacy risk. Despite such a promise, [Shariff and Sheffet, 2018] and [Vietri et al., 2020] show that standard DP is incompatible with sub-linear regret performance for contextual bandits and RL. Therefore, they embrace *joint differential privacy* (JDP) [Kearns et al., 2014], a variant of DP, ensuring that the output of all other users will not leak much information about any specific user. In some situations, they even adopt *local differential privacy* (LDP) [Duchi et al., 2013] in private RL [Garcelon et al., 2021] due to its stronger privacy guarantee, where each user's raw data must be privatized before being sent to the learning agent.

Nonetheless, private RL is still far from being well-understood. All of the previous work assumes that the losses are generated by a stochastic distribution that is stationary throughout the learning process. This assumption is quite restrictive for plenty of real-world systems since the loss function may depend on additional variables controlled by a complex and unpredictable part of the environment. These extra variables may be challenging to model and predict using a stochastic distribution and only impact the loss incurred by the user. Specifically, the loss function might unpredictably vary across episodes and even be generated by a potential adversary. In these scenarios (with privacy concerns), modeling loss functions as adversarial would be more relevant; examples include recommendation system

---

*Corresponding author

[1]We consider the setting of "losses" instead of "rewards" to be consistent with the adversarial online decision-making literature [Jain et al., 2012, Jin et al., 2020a]. One can translate between losses and rewards by simply taking negation.

[Zhou et al., 2019], medicine trials [Liu et al., 2020], and portfolio management [Luo et al., 2018]. For instance, in recommendation systems, the agent recommends items (corresponding to actions) according to users' search input (corresponding to states) and improves its performance based on users' rating (corresponding to rewards), and the rating may depend on some complex and hard-to-model historical variable of each user and reflect different preferences.

Motivated by these facts, in this paper, we focus on one fundamental model in online RL under DP constraints, i.e., private adversarial MDPs [Even-Dar et al., 2009], where the transition function is unknown and stochastic, but where the loss function can be arbitrarily determined by an oblivious adversary. To solve this problem, we are required to design private algorithms in such non-stationary environments, especially for adversarial loss functions. Moreover, we must deal with the dual complexities of adversarially changing and noisy interaction histories, which makes it challenging to utilize and generalize past experiences and adapt to evolving circumstances. To the best of our knowledge, this paper is the first to consider adversarial MDPs with both JDP and LDP guarantees. Our contributions are summarized as follows.

1. We begin with the full-information setting where the loss for *every* state-action pair is observed after each interaction. We present a general algorithm, "Private-UC-O-REPS", which uses tighter confidence bounds on components of the transition function than existing ones in the adversarial MDP literature, and enjoys refined regret bounds under JDP and LDP constraints by adopting our Central and Local Privatizer, respectively. Notably, these bounds are problem-dependent in the sense that they make appear a notion of effective support of the underlying transition function, and adapt to the difficulty of the transition dynamics. Further, they match the best bounds of non-private algorithm [Rosenberg and Mansour, 2019a] in the worst case.

2. We then consider the bandit setting where only the loss of each *visited* state-action pair is revealed after each interaction. We propose the "Private UOB-LBPS" algorithm, which involves a novel private and optimistic loss estimator, and a log-barrier regularizer for private OMD making the algorithm more stable. Meanwhile, we obtain near-optimal problem-dependent regret bounds under both JDP and LDP constraints. In particular, they also match the near-optimal regret bounds of the best non-private algorithm [Jin et al., 2020a] in the worst case.

3. We introduce novel Privatizers designed to privatize both the transition function and the adversarial losses under full-information and bandit-feedback settings. These Privatizers satisfy several key properties (see Assumptions 2.3, 3.2 and 4.1 for details), which play a critical role in the analysis to help obtain privacy guarantee and the regret bounds, and could be of interest beyond this work.

We summarize our theoretical results in Table 1. Due to space limitations, algorithms and all proof details are included in the appendix.

## 1.1 RELATED WORK

Private online decision-making in adversarial environments has been studied for over a decade, with *follow-the-leader* type algorithms commonly employed to address these challenges. Examples of such scenarios include private online convex learning [Jain et al., 2012, Agarwal et al., 2023], private expert prediction [Agarwal and Singh, 2017, Asi et al., 2023], and private (contextual) adversarial bandits [Tossou and Dimitrakakis, 2017, Agarwal and Singh, 2017, Zheng et al., 2020], etc.

Regarding private RL with regret guarantees, previous research primarily focused on MDPs in stochastic stationary environments. Notable approaches include private value-based algorithms [Vietri et al., 2020, Garcelon et al., 2021, Qiao and Wang, 2023a,b] and private policy-optimization-based algorithms [Chowdhury and Zhou, 2022a, Wu et al., 2023], particularly in tabular MDPs. Initial investigations into private linear (mixture) MDPs were also undertaken in Luyo et al. [2021], Ngo et al. [2022], Zhou [2022], Liao et al. [2023]. However, the machinery and techniques used in these papers cannot be directly applied in an adversarial environment.

Adversarial MDPs have received extensive attention, addressing non-stationary environments with both known and unknown transition functions, and considering both full-information and bandit feedback settings. While a number of algorithms with regret guarantees have been proposed for this problem recently [Rosenberg and Mansour, 2019a,b, Jin et al., 2020a, Luo et al., 2021, Zhao et al., 2023], we are not aware of any existing works on private adversarial MDPs. Thus, we believe this paper makes the first attempts at designing algorithms for adversarial MDPs with privacy and regret guarantees simultaneously.

## 2 PRELIMINARY

### 2.1 ADVERSARIAL MDPS

An episodic loop-free adversarial MDP is defined by a tuple $\left(\mathcal{X}, \mathcal{A}, P, \{\ell_k\}_{k=1}^K, H\right)$, where $\mathcal{X}, \mathcal{A}$ are state and action spaces with respective cardinalities $X$ and $A$. $P : \mathcal{X} \times \mathcal{A} \times \mathcal{X} \to [0, 1]$ is the transition function, with $P(x'|x, a)$ being the probability of transferring to state $x'$ when executing action $a$ in state $x$, and $\ell_k : \mathcal{X} \times \mathcal{A} \to [0, 1]$ is the loss function for episode $k$. Finally, $H$ denotes the length

---

[2]Under the loop-free tabular MDP in this paper, the result in episodic MDP [Jin et al., 2018] will have additional $H$ dependence.

| Feedback | Algorithm | Regret ($\epsilon$-JDP) | Regret ($\epsilon$-LDP) | Lower bound without Privacy |
|---|---|---|---|---|
| Full-info | Theorem 3.3 | $\widetilde{\mathcal{O}}\left(HC_M\sqrt{K} + \frac{X^2AH^2}{\varepsilon}\right)$ | $\widetilde{\mathcal{O}}\left(HC_M\sqrt{K} + \frac{X^2AH^2\sqrt{K}}{\varepsilon}\right)$ | $\Omega(\sqrt{XAHK})$ |
| Bandit | Theorem 4.2 | $\widetilde{\mathcal{O}}\left(HC_M\sqrt{K} + \frac{AH\sqrt{X^3K}}{\varepsilon}\right)$ | $\widetilde{\mathcal{O}}\left(HC_M\sqrt{K} + \frac{X^4AH^2\sqrt{K}}{\varepsilon}\right)$ | $\Omega\left(\sqrt{XAH^2K}\right)$ |

Table 1: Regret comparisons for private online RL on loop-free adversarial MDP[2] under both full-information and bandit settings with $\varepsilon$-JDP and $\varepsilon$-LDP guarantees. $C_M := \sum_{h=0}^{H-1}\sqrt{\sum_{(x,a)\in\mathcal{X}_h\times\mathcal{A}} C_{x,a}}$ denotes the cumulative effective support, where $C_{x,a} := [\sum_{x'\in\mathcal{X}_{h(x)+1}}\sqrt{P(x'|x,a)(1-P(x'|x,a))}]^2$ denotes the effective support of $P(\cdot|x,a)$. Finally, the lower bound follows from Jaksch et al. [2010], Jin et al. [2018]. Note that $C_M \le X\sqrt{A}$ always holds, implying that our bounds are never worse than $\widetilde{\mathcal{O}}(HX\sqrt{AK})$ of non-private setting [Rosenberg and Mansour, 2019a, Jin et al., 2020a].

of an episode. We assume that the state space $\mathcal{X}$ can be decomposed into $H+1$ non-intersecting layers $\mathcal{X}_0, \ldots, \mathcal{X}_H$ such that the first and the last layers are singletons, i.e., $\mathcal{X}_0 = \{x_0\}$ and $\mathcal{X}_H = \{x_H\}$. Furthermore, the loop-free assumption means that transitions are only possible between consecutive layers[3]. In the following, we may write $h(x)$ to refer to the index of the layer to which $x$ belongs.

The learner interacts with the MDP for $K$ episodes without knowing $P$. Before the interaction starts, an oblivious adversary selects the loss functions for all episodes $\ell_k$ arbitrarily. In episode $k$, the learner starts at state $x_0$ and decides a policy $\pi_k : \mathcal{X}\times\mathcal{A}\to[0,1]$, where we write $\pi_k(a|x)$ to denote the probability of taking action $a$ at state $x$. Then the learner executes $\pi_k$ in the MDP, generating $H$ state-action pairs $\{(x_h^k, a_h^k)\}_{h=0}^{H-1}$, where for $h\in[H-1]$[4], $a_h^k\sim\pi_k(\cdot|x_h^k)$ and $x_{h+1}^k\sim P(\cdot|x_h^k, a_h^k)$. At the end of episode $k$, the learner observes the loss feedback, which is the entire loss function $\ell_k$ under the full-information setting, or the incurred losses $\{\ell_k(x_h^k, a_h^k)\}_{h=0}^{H-1}$ under the bandit setting.

The goal of the learner is to minimize the incurred loss in $K$ episodes in expectation. More formally, for any policy $\pi$ and loss function $\ell$, we define the corresponding expected cumulative loss per episode as

$$V(\pi, \ell) = \mathbb{E}\left[\sum_{h=0}^{H-1}\ell(x_h, a_h)\Big|P, \pi\right], \qquad (2.1)$$

where the expectation is taken over trajectories $\{(x_h, a_h)\}_{h=0}^{H-1}$ generated by following $\pi$. The performance of a given learning algorithm $\mathbb{A}$ deciding $(\pi_k)_{k\ge 1}$ is measured through the notion of regret, which compares the cumulative expected loss under $\mathbb{A}$ to that incurred by the best stationary policy in hindsight, i.e.,

---

[3]This assumption – also known as layered, loop-free assumption – is a standard one in the adversarial MDP literature [Neu et al., 2012, Jin et al., 2020a].Although not necessary, it will simplify some arguments. This model is a strict generalization of the episodic setting studied in Azar et al. [2017], Jin et al. [2018], where the transitions are stationary across episode steps.

[4]For $n\in\mathbb{N}$, we define $[n] := \{1, \ldots, n\}$.

$\pi^* \in \arg\min_\pi \sum_{k=1}^K V(\pi, \ell_k)$. That is,

$$\mathcal{R}_K^{\mathbb{A}} = \sum_{k=1}^K V(\pi_k, \ell_k) - \sum_{k=1}^K V(\pi^*, \ell_k). \qquad (2.2)$$

Alternatively, the goal is to minimize the regret. By default, $\mathbb{A}$ is omitted unless explicitly noted.

### 2.1.1 Occupancy Measures

Learning in adversarial MDPs can be reformulated as an online linear optimization problem, using the notion of occupancy measures [Zimin and Neu, 2013]. Given $\pi$ and $P$, the occupancy measure $q^{P,\pi} : \mathcal{X}\times\mathcal{A}\times\mathcal{X}\to[0,1]$ is defined as follows:

$$q^{P,\pi}(x, a, x') = \mathbb{P}[x_h = x, a_h = a, x_{h+1} = x'|P, \pi],$$

where $h = h(x)$ is the index of the layer to which $x$ belongs. With slight abuse of notation, we define the probability of visiting state-action pair $(x, a)$ and that of visiting state $x$ as follows,

$$q^{P,\pi}(x, a) = \sum_{x'\in\mathcal{X}_{h(x)+1}} q^{P,\pi}(x, a, x'), \quad q^{P,\pi}(x) = \sum_{a\in\mathcal{A}} q^{P,\pi}(x, a).$$

As established in [Zimin and Neu, 2013], a valid occupancy measure $q$ satisfies: For all $h\in[H-1]$,

$$(i): \sum_{x\in\mathcal{X}_h}\sum_{a\in\mathcal{A}}\sum_{x'\in\mathcal{X}_{h+1}} q(x, a, x') = 1,$$

$$(ii): \sum_{x'\in\mathcal{X}_{h-1}}\sum_{a\in\mathcal{A}} q(x', a, x) = \sum_{x'\in\mathcal{X}_{h+1}}\sum_{a\in\mathcal{A}} q(x, a, x'), \forall x\in\mathcal{X}_h.$$

Both $(i)$ and $(ii)$ follow from the loop-free structure: $(i)$ holds since each layer is visited exactly once, whereas $(ii)$ holds due to conservation law across layers. Further, any function $q : \mathcal{X}\times\mathcal{A}\times\mathcal{X}\to[0,1]$ satisfying $(i)$-$(ii)$ induces the following transition function and policy [Rosenberg and Mansour, 2019a]:

$$P^q(x'|x, a) = \frac{q(x, a, x')}{q(x, a)}, \quad \pi^q(a|x) = \frac{q(x, a)}{q(x)}. \qquad (2.3)$$

$\Delta$ denotes the set of all valid occupancy measures. For a fixed transition function $P$, $\Delta(P)$ denotes the set of occupancy measures whose induced transition $P^q$ equals $P$. Similarly, given a set of transition functions $\mathcal{P}$, $\Delta(\mathcal{P})$ denotes the set of occupancy measures whose induced transition $P^q$ belongs to $\mathcal{P}$.

Equipped with these definitions, we can rewrite

$$V(\pi, \ell) = \sum_{h=0}^{H-1} \sum_{x \in \mathcal{X}_h} \sum_{a \in \mathcal{A}} q^{P,\pi}(x,a)\ell(x,a) = \left\langle q^{P,\pi}, \ell \right\rangle.$$

Thus, the regret of the learner can be rewritten as

$$\mathcal{R}_K^{\mathbb{A}} = \sum_{k=1}^{K} \left\langle q^{P,\pi_k} - q^*, \ell_k \right\rangle, \qquad (2.4)$$

where $q^* \in \operatorname{argmin}_{q \in \Delta(P)} \sum_{k=1}^{K} \langle q, \ell_k \rangle$ is the optimal occupancy measure in $\Delta(P)$.

When the transition function is known and the loss function is revealed at the end of each episode, this problem can be solved by an online linear optimization method [Hazan, 2016]. However, in our setting, both $P$ and $\Delta(P)$ are unknown, and we face noisy and even partial information on $\ell_k$ under the bandit setting.

## 2.2 DIFFERENTIAL PRIVACY IN ADVERSARIAL MDPS

In adversarial MDPs, each episode $k \in [K]$ can be viewed as a trajectory representing a specific user. Let $\mathcal{U}$ denote the set of all users, and let $U_K = (u_1, \ldots, u_K) \in \mathcal{U}^K$ denote a sequence of $K$ users participating in the private adversarial MDP protocol with an RL agent $\mathcal{M}$. Each user $u_k$ is identified by her interaction trajectory $S_k$, including the visited state-action pairs and the observed losses. We denote $\mathcal{M}(U_K) := \left(a_1^1, \cdots, a_H^K\right) \in \mathcal{A}^{KH}$ as the set of all actions chosen by $\mathcal{M}$ when interacting with the user sequence $U_K$, and $\mathcal{M}_{-k}(U_K) := \mathcal{M}(U_K) \setminus \left(a_h^k\right)_{h=1}^{H}$ as all the actions chosen by $\mathcal{M}$ excluding those recommended to $u_k$. Then, we first consider the notion of JDP [Kearns et al., 2014, Vietri et al., 2020].

**Definition 2.1 (Joint Differential Privacy (JDP))** *For any $\varepsilon > 0$, a mechanism $\mathcal{M} : \mathcal{U}^K \to \mathcal{A}^{KH}$ is $\varepsilon$-Joint Differentially Private ($\varepsilon$-JDP) if for all $k \in [K]$, for all user sequences $U_K, U_K' \in \mathcal{U}^K$ differing only on the $k$-th user, and for all sets of actions $\mathcal{E}_0 \subset \mathcal{A}^{(K-1)H}$,*

$$\mathbb{P}\left[\mathcal{M}_{-k}(U_K) \in \mathcal{E}_0\right] \leq \exp(\varepsilon) \cdot \mathbb{P}\left[\mathcal{M}_{-k}(U_K') \in \mathcal{E}_0\right].$$

JDP ensures that even if an adversary can observe the recommended actions to all users but $u_k$, it is still statistically difficult to identify the trajectory of $u_k$ accurately. JDP

assumes that the agent $\mathcal{M}$ is allowed to access the raw trajectories from users. However, in some scenarios, the users may not be willing to share their data with the agent directly, which motivates LDP [Duchi et al., 2013]. In this setting, the agent $\mathcal{M}$ sends policy $\pi_k$ to the user $u_k$, and the user executes $\pi_k$ and gets her trajectory $S_k$, and then privatizes it to $\widetilde{S}_k$ and sends it to $\mathcal{M}$. We denote the privacy mechanism on the users' side by $\mathcal{M}'$ and recall the definition of local differential privacy below.

**Definition 2.2 (Local Differential Privacy (LDP))** *For any $\varepsilon \geq 0$, a mechanism $\mathcal{M}'$ is $\varepsilon$-Local Differentially Private ($\varepsilon$-LDP) if for all trajectories $S, S' \in \mathcal{S}$ and for all possible subset $\mathcal{E}_0 \subset \{\mathcal{M}'(S) \,|\, S \in \mathcal{S}\}$,*

$$\mathbb{P}\left[\mathcal{M}'(S) \in \mathcal{E}_0\right] \leq \exp(\varepsilon) \cdot \mathbb{P}\left[\mathcal{M}'(S') \in \mathcal{E}_0\right],$$

*where $\mathcal{S}$ is the set of all possible trajectories.*

LDP ensures that if any adversary observes the privacy reply of user $u_k$, it is still impossible to identify her trajectory. LDP is first introduced and analyzed under RL by Garcelon et al. [2021].

We introduce some notations for later analysis. $\mathbb{I}_k\{x,a\}$ denotes an indicator function whose value is 1 if $(x,a)$ is visited in episode $k$ and 0 otherwise. Similar definition also applies to $\mathbb{I}_k\{x,a,x'\}$. Denote $S_k := \left\{\left(x_h^k, a_h^k\right)\right\}_{h=0}^{H-1} \cup \{\ell_k(x,a)\}_{(x,a)}$ and $S_k := \left\{\left(x_h^k, a_h^k, \ell_k\left(x_h^k, a_h^k\right)\right)\right\}_{h=0}^{H-1}$ as the trajectory of episode $k$ under full-information setting and bandit setting, respectively.

Under a given algorithm, denote $N_k(x,a,x') := \sum_{i=1}^{k-1} \mathbb{I}_k\{x,a,x'\}$ as the number of visits to the state-action pair $(x,a)$ followed by a visit to $x'$ *before* episode $k$, and denote $N_k(x,a) := \sum_{x' \in \mathcal{X}_{h+1}} N_k(x,a,x')$. Finally, $L_k(x,a) := \sum_{i=1}^{k-1} \ell_i(x,a)$ denotes the cumulative loss of taking action $a$ at state $x$ *before* episode $k$. In non-private learning, these counters are sufficient to find estimates of the transition function $P$ to design a policy $\pi_k$ for each episode $k$ by using model-based algorithms [Neu et al., 2012]. However, these counters are derived from the raw user trajectories, which may contain sensitive information. Therefore, we must release the counts in a privacy-preserving way, namely Privatizer, on which the learning agent would rely. Let $\widetilde{N}_k(x,a), \widetilde{N}_k(x,a,x')$, and $\widetilde{L}_k(x,a)$ denote the privatized versions of $N_k(x,a), N_k(x,a,x')$, and $L_k(x,a)$, respectively. Assumption 2.3 below requires that the private visitation counts closely approximate the true counts. The private loss for full-information and bandit settings will be specified in Section 3 and Section 4, respectively. All of the private counters will be justified by our Privatizers in Section 5.

**Assumption 2.3 (Private visitation counts)** *For any privacy budget $\varepsilon > 0$ and failure probability $\delta \in (0,1]$,*

*the private visitation counts returned by Privatizer satisfy, for some $E_{\varepsilon,\delta} > 0$, with probability at least $1 - 2\delta$, uniformly over all $(x,a,x',k)$,* $\left|\widetilde{N}_k(x,a) - N_k(x,a)\right| \le E_{\varepsilon,\delta}$, $\left|\widetilde{N}_k(x,a,x') - N_k(x,a,x')\right| \le E_{\varepsilon,\delta}$ *and* $\widetilde{N}_k(x,a) = \sum_{x' \in \mathcal{X}_{h(x)+1}} \widetilde{N}_k(x,a,x') \ge N_k(x,a)$, $\widetilde{N}_k(x,a,x') > 0$.

Using Assumption 2.3, we introduce the following private estimation of $P$ built using data available up to episode $k$:

$$\widetilde{P}_k(x'|x,a) := \frac{\widetilde{N}_k(x,a,x')}{\widetilde{N}_k(x,a)}. \tag{2.5}$$

Note that by construction of Privatizer, $\widetilde{P}_k(\cdot|x,a)$ is a valid probability distribution.

# 3 FULL-INFORMATION SETTING

Similar to the non-private algorithms [Rosenberg and Mansour, 2019a], we propose a general framework, *Private Upper Confidence Online Relative Entropy Policy Search* (Private-UC-O-REPS). The core idea is to solve an online convex optimization problem within the occupancy measure space, which combines two key elements: tighter confidence bounds on components of the transition estimate, and i.i.d. perturbations with bounded maxima on private cumulative loss functions. The complete algorithm details can be found in the appendix, and we give a brief description below.

In episode $k$, we first utilize the private counters $\widetilde{N}_{k+1}$ to establish a confidence set $\mathcal{P}_{k+1}$ that contains the true transition function with high probability, whose radius shrinks as more data is collected. Then, the occupancy measure $q_{k+1}$ is updated by solving an online optimization problem within $\mathcal{P}_{k+1}$ using the Follow-the-Regularized-Leader (FTRL) method [Hazan, 2016]. This approach is employed due to our utilization of private cumulative loss $\widetilde{L}_{k+1}$. Finally, the induced policy $\pi^{q_{k+1}}$ is chosen and executed in the next episode.

Specifically, the confidence set $\mathcal{P}_k$ is defined as

$$\mathcal{P}_k = \Big\{ P \in \triangle_{\mathcal{X} \times \mathcal{A}}^{\mathcal{X}} : \left| (P - \widetilde{P}_k)(x'|x,a) \right| \le \beta_k(x'|x,a),$$
$$\forall (x,a,x') \in \mathcal{X}_h \times \mathcal{A} \times \mathcal{X}_{h+1}, h \in [H] \Big\}, \tag{3.1}$$

where $\triangle_{\mathcal{X} \times \mathcal{A}}^{\mathcal{X}}$ denotes the set of all transition functions for the state-action space $\mathcal{X} \times \mathcal{A}$, and where the confidence width associated to $(x,a,x')$ is defined as $\beta_k(x'|x,a) = \min\left\{1, \sqrt{\frac{2\widetilde{P}_k(x'|x,a)\left(1-\widetilde{P}_k(x'|x,a)\right)\ln\iota}{\widetilde{N}_k(x,a)}} + \frac{4E_{\varepsilon,\delta}+7\ln\iota}{\widetilde{N}_k(x,a)}\right\}$, with $\iota = \frac{XAK}{\delta}$ for parameter $\delta \in (0,1)$. We have the following lemma, thanks to Bernstein-type concentration:

**Lemma 3.1** *Let $K > 0$. Then, with $P \in \mathcal{P}_k$ uniformly over all $k \in [K]$.*

Moreover, one can show that the confidence bound above is strictly tighter than those used in Rosenberg and Mansour [2019a,b], Jin et al. [2020a], which proves instrumental in obtaining problem-dependent regret bounds.

Different from the non-private adversarial MDPs, we follow the FTRL method to choose the occupancy measure $q_k$, which is a standard technique to tackle the online optimization problem, while striking a balance between exploiting past knowledge and exploring new options. Formally, given a parameter $\eta > 0$,

$$q_{k+1} = \operatorname*{argmin}_{q \in \Delta(\mathcal{P}_{k+1})} \left\langle \widetilde{L}_{k+1}, q \right\rangle + \frac{1}{\eta}\psi(q), \tag{3.2}$$

where we use the negative entropy regularizer,

$$\psi(q) = \sum_{h=0}^{H-1}\sum_{x \in \mathcal{X}_h}\sum_{a \in \mathcal{A}}\sum_{x' \in \mathcal{X}_{h+1}} q(x,a,x')\ln q(x,a,x'). \tag{3.3}$$

Note that the update can be implemented efficiently by solving an unconstrained optimization problem which has a closed-form solution, and then solving a convex projection problem which can be solved in polynomial time. (See Appendix C.2 for details.)

To privatize the loss function and achieve optimal regret guarantee in private online learning, we introduce one general assumption on the Privatizer of the loss function, which will be satisfied by our design in Section 5.

**Assumption 3.2 (Private loss in full-information setting)** *For any privacy budget $\varepsilon > 0$ and all $(x,a,k)$, $Z_k(x,a) := \widetilde{L}_k(x,a) - L_k(x,a)$ are i.i.d. random variables, satisfying $\mathbb{E}\left[\max_{x,a} Z_k(x,a) - \min_{x,a} Z_k(x,a)\right] \le E_\varepsilon$, for some $E_\varepsilon > 0$.*

Assumption 3.2 guarantees i.i.d perturbations with bounded maxima on cumulative loss, which can convert the effect of perturbed loss on regret bound to an additive and bounded bias term in the regret bound.

To provide a problem-dependent regret, we recall from [Bourel et al., 2020] the notion of *effective support*, which for a pair $(x,a)$ is defined as $C_{x,a} := \left(\sum_{x' \in \mathcal{X}_{h(x)+1}} \sqrt{P(x'|x,a)(1-P(x'|x,a))}\right)^2$. Further, the *cumulative effective support* is defined as $C_M := \sum_{h=0}^{H-1}\sqrt{\sum_{(x,a) \in \mathcal{X}_h \times \mathcal{A}} C_{x,a}}$. Both notions characterize the local structure and difficulty of the MDPs, which are always more refined than the worst case. As Bourel et al. [2020] show, $C_{x,a}$ is controlled by the number $G_{x,a}$ of successor states of $(x,a)$[5], and one has: $C_{x,a} \le G_{x,a} - 1 \le X_{h(x)+1} - 1$ and $C_M \le X\sqrt{A}$.

---

[5]For a pair $(x,a)$, we define $G_{x,a} := |\operatorname{supp}(P(\cdot|x,a))|$.

The following theorem presents a general regret bound for Private-UC-O-REPS when instantiated with any Privatizer that satisfies Assumption 2.3 and Assumption 3.2.

**Theorem 3.3** *Fix any $\varepsilon > 0$ and $K > 1$, and set $\eta = \sqrt{\frac{\ln(XA/H)}{K}}, \delta = \frac{XA}{K}$. Under Assumptions 2.3 and 3.2, the regret of Private-UC-O-REPS is*

$$\mathbb{E}\left[\mathcal{R}_K\right] \leq \widetilde{\mathcal{O}}\left(HC_M\sqrt{K} + HX^2AE_{\varepsilon,\delta} + HE_\varepsilon\right).$$

**Proof** We decompose the regret as the sum of the following two terms, $\textsc{Error} = \sum_{k=1}^K \left\langle q^{P,\pi_k} - q_k, \ell_k\right\rangle$, $\textsc{Reg} = \sum_{k=1}^K \left\langle q_k - q^*, \ell_k\right\rangle$, and then bound them separately.

$\textsc{Error}$ quantifies the cumulative difference between the loss incurred by the agent's chosen policy in the true transition $P$ and the "optimistic" MDP transition $P_k$ induced by $q_k$, where $P_k = P^{q_k} \in \mathcal{P}_k$ ensuring that $q_k = q^{P_k,\pi_k}$ by definition of $\pi_k$ and Eq. (2.3). Specifically, the agent selects occupancy measures within the confidence set, which are not exactly the occupancy measures of $P$. Since all losses are in $[0,1]$, we have $\textsc{Error} \leq \sum_{k=1}^K \sum_{x,a} \left|q^{P,\pi_k}(x,a) - q^{P_k,\pi_k}(x,a)\right|$, and with probability at least $1 - 7\delta$, $\textsc{Error} \leq \widetilde{\mathcal{O}}\left(HC_M\sqrt{K} + X^2AHE_{\varepsilon,\delta}\right)$.

The core idea of controlling $\textsc{Reg}$ is introducing a pseudo-private algorithm as an intermediate step. Instead of injecting identically distributed noise $Z_k(x,a)$ at each episode in our algorithm, the pseudo-private algorithm uses a one-shot noise injection at the very start of the algorithm, i.e., $\widetilde{L}_k(x,a) - L_k(x,a) = \widehat{Z}(x,a)$ for all $(x,a)$, and then applies the same FTRL method in Eq. (3.2) to obtain pseudo occupancy measure $\widehat{q}_k$. Benefiting from Assumption 3.2, the noise injected in both algorithms follow the same distribution, then the distribution of $q_k$ is identical to that of $\widehat{q}_k$. Therefore, we can bound $\textsc{Reg}$ by bounding the regret of the pseudo algorithm. Applying a similar FTRL analysis used in private online learning [Agarwal and Singh, 2017], the regret bound of the pseudo algorithm consists of three key components: a stability term that constrains the change in $\widehat{q}$ per episode and two bias terms arising from regularization and the one-shot noise injection. With the help of Assumption 3.2, we derive $\mathbb{E}[\textsc{Reg}] \leq \mathcal{O}\left(H\sqrt{K\ln\frac{XA}{H}} + HE_\varepsilon\right).$ ∎

## 4 BANDIT SETTING

In this section, we turn to investigate the private adversarial RL algorithm under the bandit setting. We propose *Private Upper Occupancy Bound Log-Barrier Policy Search* (Private UOB-LBPS) framework based on the non-private version in Jin et al. [2020a]. However, there are three main dif-

ferences in our algorithm. Firstly, we apply a new confidence set of the transition function defined in Eq. (3.1), which is strictly tighter and helps achieve problem-dependent regret bound. Secondly, we introduce a novel private loss estimator that maintains nice properties, i.e., optimistic estimation, bounded perturbation, and non-negativity. Thirdly, we involve a log-barrier regularizer to update occupancy measures, which helps us attain a tighter stability term.

We provide a brief description of the algorithm, deferring the full pseudo-code to the appendix. In each episode $k$, we obtain the private loss $\tilde{\ell}_k(x,a)$ for all $(x,a)$ by privatizing the observed loss with the privacy mechanism, which may be unbounded and negative. To make the loss function bounded, we require the perturbations on the observed loss not to exceed a specific threshold $E'_{\varepsilon,\delta}$ with high probability, as formally specified in Assumption 4.1. Then, we scale the private loss to $[0,1]$ to obtain

$$\ddot{\ell}_k(x,a) = \frac{\tilde{\ell}_k(x,a) + E'_{\varepsilon,\delta}}{2E'_{\varepsilon,\delta} + 1}, \qquad (4.1)$$

and then construct an optimistic loss estimators $\widehat{\ell}_k(x,a)$ using the (efficiently computable) *upper occupancy bound* $u_k$, similar to Jin et al. [2020a]:

$$\widehat{\ell}_k(x,a) = \frac{\ddot{\ell}_k(x,a)}{u_k(x,a)}, \qquad (4.2)$$

where $u_k(x,a) = \max_{P\in\mathcal{P}_k} q^{P,\pi_k}(x,a)$.

Next, we construct confidence set $\mathcal{P}_{k+1}$ in the same way as in the full-information setting (Eq. (3.1)). Finally, we find $q_{k+1}$ via Online Mirror Descent (OMD):

$$q_{k+1} = \operatorname*{argmin}_{q\in\Delta(\mathcal{P}_{k+1})} \left\langle \widehat{\ell}_k, q\right\rangle + \frac{1}{\eta}\mathcal{D}_\psi(q\|q_k), \qquad (4.3)$$

where $\mathcal{D}_\psi$ is the Bregman divergence of a log-barrier regularizer $\psi$, which leads to a better stability term in the analysis,

$$\psi(q) = \sum_{h=0}^{H-1}\sum_{x\in\mathcal{X}_h}\sum_{a\in\mathcal{A}}\sum_{x'\in\mathcal{X}_{h+1}} \log\frac{1}{q(x,a,x')}. \qquad (4.4)$$

Note that this optimization problem can also be solved efficiently via, e.g., Algorithm 4 in Lee et al. [2020].

Formally, the Privatizer for the loss function should satisfy the following assumption.

**Assumption 4.1 (Private loss in bandit feedback setting)** *For all $(x,a,k)$, $Z_k(x,a) := \tilde{\ell}_k(x,a) - \ell_k(x,a)\mathbb{I}_k(x,a)$ are i.i.d. zero-mean random variables; and for some $E'_{\varepsilon,\delta} > 0$, with probability at least $1 - \delta$ uniformly over all $(x,a,k)$, $|Z_k(x,a)| \leq E'_{\varepsilon,\delta}$.*

When instantiated with any Privatizer satisfying Assumption 2.3 and Assumption 4.1, a general regret bound for Private UOB-LBPS can be obtained as stated below.

**Theorem 4.2** *Fix any $\varepsilon > 0$ and set $\eta = \sqrt{\frac{X}{K}}, \delta = \frac{XA}{K}$. Then, under Assumption 2.3 and Assumption 4.1, the regret of Private UOB-LBPS satisfies*

$$\mathbb{E}\left[\mathcal{R}_K\right] \leq \widetilde{\mathcal{O}}\left( HC_M\sqrt{K} + HX^4 AE_{\varepsilon,\delta} + AE'_{\varepsilon,\delta}\sqrt{X^3 K} \right).$$

**Proof** We decompose the regret as the sum of the following three terms: $\text{ERROR} = \sum_{k=1}^{K}\langle q^{P,\pi_k} - q^{P_k,\pi_k}, \ell_k\rangle$, $\text{BIAS} = \sum_{k=1}^{K}\langle q^{P_k,\pi_k} - q^*, \ell_k - \widetilde{\ell}_k\rangle$, and $\text{REG} = \sum_{k=1}^{K}\langle q^{P_k,\pi_k} - q^*, \widehat{\ell}_k\rangle$.

To bound ERROR, we directly borrow the analysis in Theorem 3.3. To deal with bias caused by the scaling step, we define an intermediate variable $g_k(x,a) = \frac{\ell_k(x,a)}{2E'_{\varepsilon,\delta}+1}$, which allows for having the following decomposition:

$$\text{BIAS} = \sum_{k=1}^{K}\langle q^{P_k,\pi_k} - q^*, \ell_k - g_k\rangle + \sum_{k=1}^{K}\langle q^{P_k,\pi_k}, g_k - \widehat{\ell}_k\rangle$$
$$+ \sum_{k=1}^{K}\langle q^*, \widehat{\ell}_k - g_k\rangle.$$

The first term is bounded by $\mathcal{R}_K - \text{ERROR}$ by basic decomposition. With the help of the upper occupancy measure and the intermediate variable, the second term mainly depends on $\sum_{k=1}^{K}\sum_x |u_k(x) - q^{P,\pi_k}(x)|$, which can be nicely controlled by using our confidence set in Eq. (3.1). Besides, the third term is non-positive by the definition of our biased loss and upper occupancy measure.

Regarding REG term, benefiting from our non-negative loss estimator and the log-barrier regularizer, we have a smaller "stability" term compared with negative entropy regularizer, in the form of $\mathbb{E}[\sum_k \sum_{x,a} q_k(x,a)^2\widehat{\ell}_k^2(x,a)]$. The result comes from a standard analysis in Agarwal et al. [2017]. ∎

# 5 PRIVACY AND REGRET GUARANTEES

In this section, we design the Privatizers that satisfy the required assumptions for the considered feedback settings (full-information and bandit) and privacy constraints (JDP or LDP).

## 5.1 ACHIEVING JDP USING CENTRAL PRIVATIZER

The Central Privatizer protects the information of all individual users by privatizing all the visitation counters and losses. Specifically, given privacy budget $\varepsilon > 0$, we construct the Central Privatizer as follows:

(1) For all $(x,a,x')$, we privatize $\{N_k(x,a)\}_{k\in[K]}$ and $\{N_k(x,a,x')\}_{k\in[K]}$ by the Binary Mechanism [Chan et al.,

2011] with $\varepsilon' = \frac{\varepsilon}{3H\log K}$. Denoting the output of the Binary Mechanism by $\ddot{N}_k$, the private counts $\widetilde{N}_k$ are obtained by the procedure in Section 5.1.1.

(2) Under the full-information setting, for all $(x,a)$, we privatize $\{L_k(x,a)\}_{k\in[K]}$ by a variant of the Binary Mechanism with $\varepsilon' = \frac{\varepsilon}{3H\log K}$ (see Section 5.1.1).

(3) Under the bandit setting, for all $(k,x,a)$, we directly use the Laplace Mechanism [Dwork et al., 2014] with $\varepsilon' = \frac{\varepsilon}{3H}$, i.e., $\widetilde{\ell}_k(x,a) = \ell_k(x,a)\mathbb{I}_k(x,a) + \text{Lap}\left(\frac{3H}{\varepsilon}\right)$[6].

We summarize the properties of Central Privatizer in the following lemma.

**Lemma 5.1** *For any $\varepsilon > 0$, the Central Privatizer under both full-information and bandit settings is $\varepsilon$-DP. For any $\delta \in (0,1]$, and $K > \sqrt{XA}$, it satisfies privacy assumptions with $E_{\varepsilon,\delta} = \mathcal{O}\left(\frac{3H}{\varepsilon}\log^{1.5} K \log\iota\right)$, $E_\varepsilon = \mathcal{O}(\frac{3H}{\varepsilon}\sqrt{\log^3 K \ln(XA)})$, and $E'_{\varepsilon,\delta} = \frac{3H}{\varepsilon}\log\iota$.*

Using Lemma 5.1, as corollaries of Theorem 3.3 and Theorem 4.2, we obtain the regret and privacy guarantees for Private-UC-O-REPS and Private-UOB-LBPS instantiated using the Central Privatizer.

**Theorem 5.2 (Problem-dependent Regret under JDP)** *For any $\varepsilon > 0$, if instantiated using the Central Privatizer, Private-UC-O-REPS and Private-UOB-LBPS both satisfy $\varepsilon$-JDP. Furthermore, we obtain*

$$\mathbb{E}\left[\mathcal{R}_K^{Full}\right] \leq \widetilde{\mathcal{O}}\left(HC_M\sqrt{K} + \frac{X^2 AH^2}{\varepsilon}\right),$$
$$\mathbb{E}\left[\mathcal{R}_K^{Bandit}\right] \leq \widetilde{\mathcal{O}}\left(HC_M\sqrt{K} + \frac{AH\sqrt{X^3 K}}{\varepsilon}\right).$$

**Remark 5.3** *The Private-UC-O-REPS and Private-UOB-LBPS with JDP guarantee improve over the best existing results in non-private settings for both full-information and bandit settings [Rosenberg and Mansour, 2019a, Jin et al., 2020a] by making appear a problem-dependent term and also match them in the worst case, i.e., $\widetilde{\mathcal{O}}\left(HX\sqrt{AK}\right)$.*

**Remark 5.4** *Compared to the lower bound for the stochastic RL with the JDP guarantee in Vietri et al. [2020], $\Omega\left(H\sqrt{XAK} + \frac{XAH\log K}{\epsilon}\right)$, our bounds have an optimal dependency on the privacy budget $\varepsilon$. In terms of $K$, the privacy cost in the full-information setting is a lower order term compared with the non-private term, which is dominated by the estimation error on the transition function due to private visitation counters. However, the privacy cost is*

---

[6]Here, we slightly overloaded notation, and used $\text{Lap}(\cdot)$ to represent a zero-mean Laplace variable with parameter $\cdot$.

*sub-optimal in the bandit setting, and the dominant factor is regret associated with the private loss estimator, given the stronger privacy guarantee for loss in Lemma 5.6. This gap may be attributed to the inefficiency of our privacy mechanism but might also arise due to a loose lower bound.*

### 5.1.1 Post-processing steps

During the $k$-th episode, given the noisy counts $\ddot{N}_k(x,a)$, $\ddot{N}_k(x,a,x')$, $\ddot{L}_k(x,a)$ for all $(x,a,x')$ from the classical Binary Mechanism [Chan et al., 2011], we construct the following private counters as follows.

**Private visitation counters.** To satisfy Assumption 2.3, we use the techniques from Qiao and Wang [2023a]. Firstly, we solve the optimization problem[7] for all $(x,a)$ below.

$$\min t \text{ s.t. } n(x') \geq 0, \quad \forall x',$$
$$\left| n(x') - \ddot{N}_k(x,a,x') \right| \leq t, \quad \forall x',$$
$$\left| \sum_{x' \in \mathcal{X}_{h(x)+1}} n(x') - \ddot{N}_k(x,a) \right| \leq \frac{E_{\varepsilon,\delta}}{4}. \tag{5.1}$$

Letting $\bar{N}_k(x,a,x')$ denote a minimizer of this problem, we define $\bar{N}_k(x,a) = \sum_{x' \in \mathcal{X}_{h(x)+1}} \bar{N}_k(x,a,x')$. By adding an additional term, as done below, we make sure that the private counts $\widetilde{N}_k(x,a)$ never underestimate the respective true counts:

$$\widetilde{N}_k(x,a) = \bar{N}_k(x,a) + \frac{E_{\varepsilon,\delta}}{2},$$
$$\widetilde{N}_k(x,a,x') = \bar{N}_k(x,a,x') + \frac{E_{\varepsilon,\delta}}{2X_{h+1}}. \tag{5.2}$$

The private counts $\widetilde{N}_k$ satisfy the following property.

**Lemma 5.5** *Suppose $\ddot{N}_h^k$ satisfy*

$$\left| \ddot{N}_k(x,a,x') - N_k(x,a,x') \right| \leq \frac{E_{\varepsilon,\delta}}{4},$$
$$\left| \ddot{N}_k(x,a) - N_k(x,a) \right| \leq \frac{E_{\varepsilon,\delta}}{4}, \tag{5.3}$$

*for all $(h,k,x,a,x')$, with probability $1 - 2\delta$. Then, $\widetilde{N}_k$ derived from Eq. (5.1) and Eq. (5.2) satisfy Assumption 2.3.*

**Private loss in full-information setting.** We use a variant of the Binary Mechanism which maintains the same privacy guarantee as the standard Binary Mechanism but has better distributional properties for our problem (see Lemma F.1). That is, for all $(k,x,a)$, we post-process $\ddot{L}_k(x,a)$ by injecting more noise such that the perturbation on $L_k(x,a)$

---

[7]Note that Problem (5.1) is a linear program with $\mathcal{O}(X_{h(x)+1})$ variables and $\mathcal{O}(X_{h(x)+1})$ linear constraints, which can be solved efficiently using existing algorithms for linear programming. A fast implementation could be via the simplex method [Ficken, 2015].

is a summation of $\lceil \log K \rceil$ i.i.d. Laplace variables. Thus, Assumption 3.2 is satisfied by the maxima of the sum of i.i.d. Laplace variables (see Lemma E.7).

**Private loss in bandit setting.** Assumption 4.1 is satisfied by the concentration of Laplace variables [Boucheron et al., 2003]. Moreover, the following lemma for private loss is also held by using the property of the Laplace Mechanism.

**Lemma 5.6** *As defined in Section 5.1, the sequence $\left\{ \tilde{\ell}_k(x,a) \right\}_{(x,a,k)}$ satisfies both $\varepsilon/3$-DP and $\varepsilon/3$-LDP.*

### 5.2 ACHIEVING LDP USING LOCAL PRIVATIZER

The Local-Privatizer, at each episode $k$, releases the private counts by perturbing the statistics computed from the trajectory generated in that episode. Given the privacy budget $\varepsilon > 0$, we construct Local Privatizer as follows:

(1) For all $(k,x,a,x')$, we perturb the true count $\sigma_k(x,a) := \mathbb{I}_k(x,a)$ by injecting independent Laplace noises: $\widetilde{\sigma}_k(x,a) = \sigma_k(x,a) + \text{Lap}(3H/\varepsilon)$. Then, the noisy counts are calculated by $\ddot{N}_k(x,a) = \sum_{i=1}^{k-1} \widetilde{\sigma}_i(x,a)$. The counter $\ddot{N}_k(x,a,x')$ is obtained in a similar way. To this end, through the post-processing in Section 5.1.1, we get the private counts $\widetilde{N}_k$.

(2) Under the full-information setting, for all $(k,x,a,x')$, we perturb the observed loss by adding independent Laplace noise: $\tilde{\ell}_k(x,a) = \ell_k(x,a) + \text{Lap}(3H/\varepsilon)$. The accumulative statistic is calculated by $\widetilde{L}_k(x,a) = \sum_{i=1}^{k-1} \tilde{\ell}_k(x,a)$.

(3) Under the bandit setting, we apply the same mechanism as in Section 5.1, with the help of Lemma 5.6.

The properties of the Local-Privatizer are as follows.

**Lemma 5.7** *For any $\varepsilon > 0$, the Local Privatizer under both full-information and bandit settings is $\varepsilon$-LDP. For any $\delta \in (0,1]$, and $K > \ln(XA)/2$, it satisfies privacy assumptions with $E_{\varepsilon,\delta} = \mathcal{O}\left( \frac{3H}{\varepsilon} \sqrt{K \log \iota} \right)$, $E_{\varepsilon} = \mathcal{O}\left( \frac{3H}{\varepsilon} \sqrt{K \ln(XA)} \right)$, and $E'_{\varepsilon,\delta} = \frac{3H}{\varepsilon} \log \iota$.*

Combining Lemma 5.7, Theorem 3.3, and Theorem 4.2, we obtain the following regret bound:

**Theorem 5.8 (Problem-dependent Regret under LDP)**
*For any $\varepsilon > 0$, if instantiated using the Local Privatizer, Private-UC-O-REPS and Private-UOB-LBPS both satisfy $\varepsilon$-LDP. Furthermore, we obtain the expected regret,*

$$\mathbb{E}\left[ \mathcal{R}_K^{Full} \right] \leq \widetilde{\mathcal{O}}\left( HC_M\sqrt{K} + \frac{X^2AH^2\sqrt{K}}{\varepsilon} \right),$$

$$\mathbb{E}\left[ \mathcal{R}_K^{Bandit} \right] \leq \widetilde{\mathcal{O}}\left( HC_M\sqrt{K} + \frac{X^4AH^2\sqrt{K}}{\varepsilon} \right).$$

**Remark 5.9** *Similar to the JDP case, the Private-UC-O-REPS and Private-UOB-LBPS with LDP guarantee also enjoy the problem-dependence efficiency, and match the best regret bounds in non-private settings for both full-information and bandit settings [Rosenberg and Mansour, 2019a, Jin et al., 2020a] in the worst case.*

**Remark 5.10** *In the case of LDP, Garcelon et al. [2021] implies a lower bound of $\Omega\left(\frac{H\sqrt{XAK}}{\varepsilon}\right)$ for the stochastic episodic RL, for the privacy-related term assuming small enough $\epsilon$ (corresponding to high privacy regime). Our bounds also have an optimal dependency on privacy budget $\varepsilon$ and episode number $K$, but a worse dependency on the size of the size of the state-space. In the full-information setting, this gap is mainly due to the $L_1$-norm estimated error on the transition function. In comparison, in the bandit setting, the main factor is the bias between the upper occupancy measure and the true occupancy measure, influenced by our component-wise confidence set.*

### 5.3 FURTHER DISCUSSIONS

Our Privatizer for visitation counters in Assumption 2.3 is the same as the previous work [Qiao and Wang, 2023a], but the motivation is different. In our setting, we apply the post-processing step for $\ddot{N}_k$ to ensure that $\widetilde{P}_k$ is a valid probability distribution so that we can construct a valid occupancy measure space for online optimization (Eq. (3.2) and Eq. (4.3)). Meanwhile, the novel Assumption 3.2 for private cumulative loss $\widetilde{L}_k$ helps separate the impact of noise on regret for online optimization. The Assumption 4.1 for private loss estimators also bridges the privacy protection between DP and LDP and plays a vital role in the regret minimization procedure.

The Laplace noise involved in our Privatizer can also be replaced with other noises like Gaussian noise [Dwork et al., 2014]. According to Theorem 3.3 and Theorem 4.2, the regret bounds can be easily derived by plugging in the corresponding precision level $E_{\varepsilon,\delta}$, $E_\varepsilon$, and $E'_{\varepsilon,\delta}$.

## 6 CONCLUSION AND FUTURE WORK

In this paper, we presented the first differentially private algorithms for adversarial MDPs with unknown transitions under both full information and bandit settings. Our designs rely on tighter confidence bounds on the components of the transition function, novel central and local Privatizers for transition functions, and adversarial losses separately. By instantiating the proposed Privatizers, both algorithms are proven to achieve near-optimal problem-dependent regret bounds, satisfying JDP or LDP privacy guarantees. Further, the bounds also match non-private state-of-the-art bounds in the worst case.

A natural direction of future work is to close the gap between the upper and lower bounds on regret. A similar gap remains open *without* privacy considerations, which requires new progress in the non-private setting. We believe designing refined privacy mechanisms for adversarial MDPs or establishing a lower bound here also leads to interesting technical questions in this domain.

Considering that occupancy-measure-based algorithms are computationally intensive, it will be a promising direction to privatize policy-optimization-based algorithms, e.g., Luo et al. [2021], Dann et al. [2023]. Besides, considering MDPs with function approximation [Jin et al., 2020b, Sherman et al., 2023] also has considerable potential for real-world applications.

### Acknowledgements

We thank the anonymous reviewers and area chair for their valuable comments, and the helpful discussion with Chloé Rouyer, Yi-Shan Wu. This work is supported by National Natural Science Foundation of China (NSFC) under Grant 62293511, 62103371, 62273305; Fundamental Research Funds for the Central Universities (FRF) 226-2023-00111, 226-2024-00004; Zhejiang Provincial Natural Science Foundation under Grant (ZPNSFC) LZ23F030009, LZ22F030010; Shanghai Pujiang Program under grant 22PJ1404900; and the Independent Research Fund Denmark under grant 1026-00397B.

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

# Differentially Private No-regret Exploration in Adversarial Markov Decision Processes (Supplementary Material)

**Shaojie Bai**[1,3]    **Lanting Zeng**[1]    **Chengcheng Zhao**[1]    **Xiaoming Duan**[2]    **Mohammad Sadegh Talebi**[3]

**Peng Cheng** [††1]                                **Jiming Chen**[1]

[1]Zhejiang University, Hangzhou, China
[2]Shanghai Jiaotong University, Shanghai, China
[3]University of Copenhagen, Copenhagen, Denmark

## A    EXTENDED RELATED WORK

**Regret minimization for adversarial MDPs.** Over the past decade, research on adversarial (tabular) Markov Decision Processes has covered various scenarios, including known and unknown transition functions, as well as full-information and bandit feedback settings. In a scenario with a known transition function, Zimin and Neu [2013] introduces the O-REPS algorithm, which employs Online Mirror Descent over the space of occupancy measures. This approach yields regret bounds of $\widetilde{O}(H\sqrt{K})$. To tackle the challenges posed by unknown transition functions, Rosenberg and Mansour [2019a] combine confidence sets and Online Mirror Descent. This hybrid approach achieves the best-known regret bound of $\widetilde{O}(\sqrt{X^2AH^2K})$ under the full-information setting. In the bandit setting, Rosenberg and Mansour [2019b] develop inverse importance-weighted loss estimators and obtain regret bounds of $\widetilde{O}(\frac{\sqrt{X^2AH^2K}}{\alpha})$ under the $\alpha$-reachability assumption. Building on these works, Jin et al. [2020a] further improves the regret bounds by introducing biased and optimistic loss estimators along with a tighter confidence set. These advancements lead to the best regret bound of $\widetilde{O}(\sqrt{X^2AH^2K})$. Additionally, policy-optimization-based methods have been developed by Shani et al. [2020] and Luo et al. [2021]. In particular, [Luo et al., 2021] matches the best regret bound achieved through the OMD methods.

**Private online learning.** Private online learning has been a subject of extensive research for over a decade, and *follow-the-leader* type algorithms have been employed in various scenarios. For instance, Guha Thakurta and Smith [2013] introduce a private *follow-the-approximate-leader* method for online convex learning. Additionally, Agarwal and Singh [2017] and Kairouz et al. [2021] propose private *follow-the-regularized-leader* algorithms for online linear optimization and online federated learning, respectively. In the context of private bandit learning, Tossou and Dimitrakakis [2017] design a private variant of the EXP3 algorithm for adversarial bandits, while Agarwal and Singh [2017] and Zheng et al. [2020] explore private adversarial linear bandits and private convex bandit learning. Their works offered general reduction frameworks that could achieve nearly optimal regret. Furthermore, a substantial body of research has focused on private online learning with contextual information, spanning areas like private contextual bandit problems Shariff and Sheffet [2018], Zheng et al. [2020], Chowdhury and Zhou [2022b], Charisopoulos et al. [2023], as well as private stochastic reinforcement learning Vietri et al. [2020], Garcelon et al. [2021], Qiao and Wang [2023a], Liao et al. [2023]. Notably, despite the advancements in private online learning, none of the existing work has addressed the specific challenges posed by private reinforcement learning in the context of Adversarial Markov Decision Processes. This uncharted territory introduces the unique challenge of private online learning within an adversarial environment characterized by contextual dynamics.

---

[*]Corresponding author
[††]Corresponding author

# B TABLE OF NOTATION

| Symbol | Descriptions |
|---|---|
| $\mathcal{X}$ | state space with cardinality $X$ |
| $\mathcal{A}$ | action space with cardinality $A$ |
| $P$ | transition function |
| $K$ | number of episodes |
| $H$ | episode length |
| $\Delta(P)$ | the set of occupancy measure induced by transition $P$ |
| $\mathcal{P}$ | confidence set of transition function |
| $\pi_k$ | policy at episode $k$ |
| $q^{P,\pi_k}$ | occupancy measure of policy $\pi_k$ under transition $P$ |
| $P_k$ | transition function induced by occupancy measure $q_k$ at episode $k$ |
| $q^{P_k,\pi_k}$ | occupancy measure of policy $\pi_k$ under transition $P_k$, i.e., $q_k$ |
| $u_k$ | upper occupancy measure within $\Delta(\mathcal{P}_k)$ |
| $\pi^*$ | optimal policy |
| $q^*$ | optimal occupancy measure |
| $\varepsilon$ | privacy budget |
| $\delta$ | failure probability |
| $\eta$ | learning rate for online learning algorithms |
| $\psi(q)$ | regularizer function of $q$ for online learning algorithm |
| $N_k(x,a)$ | count of visiting state-action pair $(x,a)$ *before* episode $k$ |
| $N_k(x,a,x')$ | count of going to state $x'$ from $x$ upon playing action $a$ *before* episode $k$ |
| $\ell_k(x,a)$ | loss of the state-action pair $(x,a)$ at episode $k$ |
| $L_k(x,a)$ | cumulative loss of the state-action pair $(x,a)$ *before* episode $k$ |
| $\widetilde{N}_k(x,a)$ | the privatized version of $N_k(x,a)$ |
| $\widetilde{N}_k(x,a,x')$ | the privatized version of $N_k(x,a,x')$ |
| $\widehat{N}_k(x,a)$ | an optimistic value defined over $\widetilde{N}_k(x,a)$ in Eq.D.1 |
| $\widetilde{L}_k(x,a)$ | the privatized version of $L_k(x,a)$ |
| $\tilde{\ell}_k(x,a)$ | the privatized version of observed loss $\ell_k(x,a)\,\mathbb{I}_k(x,a)$ |
| $\ddot{\ell}_k(x,a)$ | the scaled version of private loss $\tilde{\ell}_k(x,a)$ |
| $\widehat{\ell}_k(x,a)$ | the final loss estimator of $(x,a)$ |
| $\bar{P}_k$ | transition function estimated by using true counts at episode $k$ |
| $\widetilde{P}_k$ | transition function estimated by using private counts at episode $k$ |
| $\beta_k$ | confidence width for the element of private transition estimation at episode $k$ |
| $E_{\varepsilon,\delta}$ | precision level for visitation counters |
| $E_\varepsilon$ | precision level for loss counter in full-information setting |
| $E'_{\varepsilon,\delta}$ | precision level for loss estimator in bandit-feedback setting |

Table 2: List of Notation

# C   OMITTED DETAILS FOR THE ALGORITHM

## C.1   PRIVATE UC-O-REPS ALGORITHM

---

**Algorithm 1** Private UC-O-REPS

---

**Parameters:** state space $\mathcal{X}$, action space $\mathcal{A}$, episode number $K$, episode length $H$, learning rate $\eta > 0$, confidence parameter $\delta \in (0, 1]$, privacy budget $\varepsilon > 0$ and a Privatizer

**Initialization:**

Initialize confidence set $\mathcal{P}_1$ as the set of all transitions.

For all $(x, a, x') \in \mathcal{X} \times \mathcal{A} \times \mathcal{X}_{h(x)+1}$, initialize private counts, $\widetilde{N}_1(x, a, x') = \widetilde{N}_1(x, a) = \widetilde{L}_1(x, a) = 0$, and occupancy measure, $q_1(x, a, x') = \frac{1}{X_{h(x)} A X_{hx+1}}$.

Initialize policy $\pi_1 = \pi^{q_1}$.

Set precision levels $E_{\varepsilon, \delta}$ and $E_\varepsilon$ for Privatizer.

1: **for** $k = 1$ to $K$ **do**
2:      Execute policy $\pi_k$ for $H$ steps and obtain interaction history $\left\{ \left( x_h^k, a_h^k \right) \right\}_{h=0}^{H-1}$, and observe loss function $\ell_k$.
3:      Recieve private counts $\widetilde{L}_{k+1}(x, a)$, $\widetilde{N}_{k+1}(x, a)$, $\widetilde{N}_{k+1}(x, a, x')$ from Privatizer.
4:      Compute private transition estimate, for all $(x, a, x') \in \mathcal{X} \times \mathcal{A} \times \mathcal{X}_{h(x)+1}$,

$$\widetilde{P}_{k+1}(x'|x, a) = \frac{\widetilde{N}_{k+1}(x, a, x')}{\widetilde{N}_{k+1}(x, a)}.$$

5:      Update the confidence set $\mathcal{P}_{k+1}$ based Equation (3.1).

$$\mathcal{P}_{k+1} = \left\{ P : \left| P(x'|x, a) - \widetilde{P}_{k+1}(x'|x, a) \right| \leq \beta_{k+1}(x'|x, a), \quad \forall (x, a, x') \in \mathcal{X}_h \times \mathcal{A} \times \mathcal{X}_{h+1}, h \in [H] \right\}.$$

6:      Update the occupancy measure

$$q_{k+1} = \underset{q \in \Delta(\mathcal{P}_{k+1})}{\arg\min} \left\langle \widetilde{L}_{k+1}, q \right\rangle + \frac{1}{\eta} \psi(q),$$

     where $\psi(q) = \sum_{(x,a,x') \in \mathcal{X} \times \mathcal{A} \times \mathcal{X}} q(x, a, x') \ln q(x, a, x')$.

7:      Update the policy

$$\pi_{k+1} = \pi^{q_{k+1}}.$$

8: **end for**

---

## C.2 UPDATING OCCUPANCY MEASURE EFFICIENTLY FOR PRIVATE-UC-O-REPS

This subsection explains how to implement the update defined in (3.2) for Algorithm 1, Private-UC-O-REPS, efficiently.

Similar to the approaches in Neu et al. [2012], Rosenberg and Mansour [2019a], Jin et al. [2020a], we provide details of the modification here for completeness. Based on the property of FTRL, this optimization can be reformulated as first solving the unconstrained optimization problem,

$$q'_{k+1} = \underset{q}{\mathrm{argmin}} \left\langle \widetilde{L}_{k+1}, q \right\rangle + \frac{1}{\eta} \psi(q), \tag{C.1}$$

and then projecting the result

$$q_{k+1} = \underset{q \in \Delta(\mathcal{P}_{k+1})}{\mathrm{argmin}} \mathcal{D}_\psi(q \| q'_{k+1}), \tag{C.2}$$

where $\psi(q)$ is negative entropy regularizer function defined in Eq. (3.3), and $\mathcal{D}_\psi(q \| q') = \sum_{(x,a,x') \in \mathcal{X} \times \mathcal{A} \times \mathcal{X}_{h(x)+1}} \left( q(x,a,x') \ln \frac{q(x,a,,x')}{q'(x,a,x')} - (q(x,a,x') - q'(x,a,x')) \right)$ is the corresponding Bregman divergence. For the unconstrained optimization, we have the optimal solution directly through the Lagrange method,
$q'_{k+1}(x,a,x') = \frac{\exp\left(-\eta \widetilde{L}_{k+1}(x,a)\right)}{\sum_{(x,a,x') \in \mathcal{X}_h \times \mathcal{A} \times \mathcal{X}_{h+1}} \left(\exp\left(-\eta \widetilde{L}_{k+1}(x,a)\right)\right)}$ for all $(x,a,x') \in \mathcal{X} \times \mathcal{A} \times \mathcal{X}_{h(x)+1}$.

For the second step, we can rewrite this constrained optimization problem with the following set of linear equations, which can be solved in polynomial time.

$$\min_{q} \quad \mathcal{D}_\psi(q \| q'_{k+1})$$

$$\begin{aligned}
\text{s.t.} \quad & \sum_{(x,a,x') \in \mathcal{X}_h \times \mathcal{A} \times \mathcal{X}_{h+1}} q(x,a,x') = 1 && \forall h \\
& \sum_{a \in \mathcal{A}, x' \in \mathcal{X}_{h+1}} q(x,a,x') = \sum_{x' \in \mathcal{X}_{h-1}, a \in \mathcal{A}} q(x',a,x) && \forall h, \forall x \in \mathcal{X}_h \\
& \left(\widetilde{P}_{k+1}(x'|x,a) + \beta_{k+1}(x,a,x')\right) \cdot \sum_{x' \in \mathcal{X}_{h+1}} q(x,a,x') \geq q(x,a,x') && \forall h, \forall (x,a,x') \in \mathcal{X}_h \times \mathcal{A} \times \mathcal{X}_{h+1} \\
& \left(\widetilde{P}_{k+1}(x'|x,a) - \beta_{k+1}(x,a,x')\right) \cdot \sum_{x' \in \mathcal{X}_{h+1}} q(x,a,x') \leq q(x,a,x') && \forall h, \forall (x,a,x') \in \mathcal{X}_h \times \mathcal{A} \times \mathcal{X}_{h+1} \\
& q(x,a,x') \geq 0 && \forall h, \forall (x,a,x') \in \mathcal{X}_h \times \mathcal{A} \times \mathcal{X}_{h+1}
\end{aligned}$$

This problem can be further reformulated into a dual problem, which is a convex optimization problem with only non-negativity constraints and thus can be solved more efficiently.

**Lemma C.1** *The dual problem of* (C.2) *is*

$$\mu_k^+, \mu_k^-, \nu_k = \underset{\mu^+, \mu^-, \nu \geq 0}{\mathrm{argmin}} \sum_{h=0}^{H-1} \ln \left( \sum_{(x,a,x') \in \mathcal{X}_h \times \mathcal{A} \times \mathcal{X}_{h+1}} S_{k,h}^{\mu^+, \mu^-, \nu}(x,a,x') \right), \tag{C.3}$$

*where $\mu^+ := \{\mu^+(x,a,x')\}_{(x,a,x')}, \mu^- := \{\mu^-(x,a,x')\}_{(x,a,x')}$ and $\nu := \{\nu(x)\}_x$ are dual variables and*

$$S_{k,h}^{\mu^+, \mu^-, \nu}(x,a,x') = \frac{1}{\sum_{(x,a,x') \in \mathcal{X}_h \times \mathcal{A} \times \mathcal{X}_{h+1}} \exp\left(-\eta \widetilde{L}_{k+1}(x,a)\right)} \exp\left(-\eta \widetilde{L}_{k+1}(x,a) + B_{k,h}^{\mu^+, \mu^-, \nu}(x,a,x')\right),$$

$$B_{k,h}^{\mu^+, \mu^-, \nu}(x,a,x') = -\nu(x) + \nu(x') - \mu^+(x,a,x') + \mu^-(x,a,x')$$
$$+ \sum_{y \in \mathcal{X}_{h(x)+1}} \left[ \widetilde{P}_{k+1}(y|x,a) \left(\mu^+(x,a,y) - \mu^-(x,a,y)\right) + \beta_{k+1}(x,a,y) \left(\mu^+(x,a,y) + \mu^-(x,a,y)\right) \right].$$

*Furthermore, the optimal solution to this projection is given by, for any $(x,a,x') \in \mathcal{X}_h \times \mathcal{A} \times \mathcal{X}_{h+1}$,*

$$q_{k+1}(x,a,x') = \frac{S_{k,h}^{\mu^+, \mu^-, \nu}(x,a,x')}{\sum_{(x,a,x') \in \mathcal{X}_h \times \mathcal{A} \times \mathcal{X}_{h+1}} S_{k,h}^{\mu^+, \mu^-, \nu}(x,a,x')}. \tag{C.4}$$

**Proof** In the following proof, we omit the non-negativity constraints of Problem (C.3). This is without loss of generality since the optimal solution for the modified version without non-negativity constraints turns out to always satisfy the non-negativity constraints. We write the Lagrangian as,

$$
\mathcal{L}(q, \lambda, \nu, \mu^+, \mu^-) = \mathcal{D}_\psi(q\|q'_{k+1}) + \sum_{h=0}^{H-1} \lambda_h \left( \sum_{(x,a,x')\in\mathcal{X}_h\times\mathcal{A}\times\mathcal{X}_{h+1}} q(x,a,x') - 1 \right)
$$

$$
+ \sum_{h=1}^{H-1} \sum_{x\in\mathcal{X}_h} \nu(x) \left( \sum_{a\in\mathcal{A},x'\in\mathcal{X}_{h+1}} q(x,a,x') - \sum_{x'\in\mathcal{X}_{h-1},a\in\mathcal{A}} q(x',a,x) \right)
$$

$$
+ \sum_{h=0}^{H-1} \sum_{(x,a,x')\in\mathcal{X}_h\times\mathcal{A}\times\mathcal{X}_{h+1}} \mu^+(x,a,x') \left( q(x,a,x') - \left( \widetilde{P}_{k+1}(x'|x,a) + \beta_{k+1}(x,a,x') \right) \cdot \sum_{y\in\mathcal{X}_{h+1}} q(x,a,y) \right)
$$

$$
+ \sum_{h=0}^{H-1} \sum_{(x,a,x')\in\mathcal{X}_h\times\mathcal{A}\times\mathcal{X}_{h+1}} \mu^-(x,a,x') \left( \left( \widetilde{P}_{k+1}(x'|x,a) - \beta_{k+1}(x,a,x') \right) \cdot \sum_{y\in\mathcal{X}_{h+1}} q(x,a,y) - q(x,a,x') \right)
$$

where $\lambda := \{\lambda_h\}_h$, $\nu := \nu(x)_x$ and $\mu := \{\mu^+(x,a,x'), \mu^-(x,a,x')\}_{(x,a,x')}$ are Lagrange multipliers. We denote $\nu(x_0) = \nu(x_K) = 0$ to avoid addressing the edge cases explicitly. Now, we consider the derivative with respect to $q(x,a,x')$.

$$
\frac{\partial\mathcal{L}}{\partial q(x,a,x')} = \ln q(x,a,x') - \ln q'(x,a,x') + \lambda_h + \nu(x) - \nu(x') + \mu^+(x,a,x') - \mu^-(x,a,x')
$$

$$
- \sum_{y\in\mathcal{X}_{h(x)+1}} \left[ \widetilde{P}_{k+1}(y|x,a) \left( \mu^+(x,a,y) - \mu^-(x,a,y) \right) + \beta_{k+1}(x,a,y) \left( \mu^+(x,a,y) + \mu^-(x,a,y) \right) \right]
$$

$$
= \ln q(x,a,x') - \ln q'(x,a,x') + \lambda_h - B_{k,h}^{\mu^+,\mu^-,\nu}(x,a,x').
$$

Setting the gradient to zero and using the explicit form of $q'_{k+1}(x,a,x')$ we obtain,

$$
q_{k+1}(x,a,x') = q'_{k+1}(x,a,x') \cdot \exp\left( -\lambda_h + B_{k,h}^{\mu^+,\mu^-,\nu}(x,a,x') \right)
$$

$$
= \frac{1}{\sum_{(x,a,x')\in\mathcal{X}_h\times\mathcal{A}\times\mathcal{X}_{h+1}} \exp\left( -\eta\widetilde{L}_{k+1}(x,a) \right)} \exp\left( -\eta\widetilde{L}_{k+1}(x,a) - \lambda_h + B_{k,h}^{\mu^+,\mu^-,\nu}(x,a,x') \right).
$$

Using the first constraint $\sum_{(x,a,x')\in\mathcal{X}_h\times\mathcal{A}\times\mathcal{X}_{h+1}} q(x,a,x') = 1$ to discover that

$$
e^{\lambda_h} = \sum_{(x,a,x')\in\mathcal{X}_h\times\mathcal{A}\times\mathcal{X}_{h+1}} \frac{1}{\sum_{(x,a,x')\in\mathcal{X}_h\times\mathcal{A}\times\mathcal{X}_{h+1}} \exp\left( -\eta\widetilde{L}_{k+1}(x,a) \right)} \exp\left( -\eta\widetilde{L}_{k+1}(x,a) + B_{k,h}^{\mu^+,\mu^-,\nu}(x,a,x') \right)
$$

$$
= \sum_{(x,a,x')\in\mathcal{X}_h\times\mathcal{A}\times\mathcal{X}_{h+1}} S_{k,h}^{\mu^+,\mu^-,\nu}(x,a,x').
$$

Taking $\lambda_h$ back, we obtain the explicit form of the solution,

$$
q_{k+1}(x,a,x') = \frac{S_{k,h}^{\mu^+,\mu^-,\nu}(x,a,x')}{\sum_{(x,a,x')\in\mathcal{X}_h\times\mathcal{A}\times\mathcal{X}_{h+1}} S_{k,h}^{\mu^+,\mu^-,\nu}(x,a,x')}.
$$

We note the equivalent formula of Lagrangian,

$$
\mathcal{L}(q, \lambda, \nu, \mu^+, \mu^-\mu) = \sum_{h=0}^{H-1} \sum_{(x,a,x')\in\mathcal{X}_h\times\mathcal{A}\times\mathcal{X}_{h+1}} \left( \left( \frac{\partial\mathcal{L}}{\partial q(x,a,x')} - 1 \right) q(x,a,x') + q'_{k+1}(x,a,x') \right) - \sum_{h=1}^{H-1} \lambda_h.
$$

It is straightforward to check that strong duality holds, and thus the optimal dual variables of $\mu^+, \mu^-, \nu$ are given by

$$
\mu^{+*}, \mu^{-*}, \nu^* = \underset{\mu^+,\mu^-,\nu}{\arg\max} \max_\lambda \min_q \mathcal{L}(q, \lambda, \nu, \mu^+, \mu^-) = \underset{\mu^+,\mu^-,\nu}{\arg\max} \mathcal{L}(q^*, \lambda^*, \nu, \mu^+, \mu^-) \tag{C.5}
$$

$$= \operatorname*{argmax}_{\mu^+,\mu^-,\nu} \left( -H + \sum_{h=0}^{H-1} \sum_{(x,a,x') \in \mathcal{X}_h \times \mathcal{A} \times \mathcal{X}_{h+1}} q'_{k+1}(x,a,x') - \sum_{h=0}^{H-1} \lambda_h \right) \qquad \text{(C.6)}$$

$$= \operatorname*{argmin}_{\mu^+,\mu^-,\nu} \sum_{h=0}^{H-1} \lambda_h = \operatorname*{argmin}_{\mu^+,\mu^-,\nu} \sum_{h=0}^{H-1} \ln \left( \sum_{(x,a,x') \in \mathcal{X}_h \times \mathcal{A} \times \mathcal{X}_{h+1}} S_{k,h}^{\mu^+,\mu^-,\nu}(x,a,x') \right). \qquad \text{(C.7)}$$

We apply $\frac{\partial \mathcal{L}}{\partial q(x,a,x')} = 0$ to Eq. (C.6), and in Eq. (C.7) the first two terms are independent of dual variables. Thus, combing all equations for $q_{k+1}, \lambda^*, \mu^{+*}, \mu^{-*}, \nu^*$ finishes the proof. ∎

The pseudo-code of this efficient algorithm for Private-UC-O-REPS is as follows.

---

**Algorithm 2** Updating Occupancy Measure and Policy Procedure

---

**Input:** transition function estimate $\widetilde{P}_{k+1}$, cumulative private loss function $\widetilde{L}_{k+1}$

1: Solve optimization problem C.3

$$\mu_k^+, \mu_k^-, \nu_k = \operatorname*{argmin}_{\mu^+,\mu^-,\nu \geq 0} \sum_{h=0}^{H-1} \ln \left( \sum_{(x,a,x') \in \mathcal{X}_h \times \mathcal{A} \times \mathcal{X}_{h+1}} S_{k,h}^{\mu^+,\mu^-,\nu}(x,a,x') \right),$$

where $\mu^+ := \{\mu^+(x,a,x')\}_{(x,a,x')}$, $\mu^- := \{\mu^-(x,a,x')\}_{(x,a,x')}$ and $\nu := \{\nu(x)\}_x$, and $S_{k,h}^{\mu^+,\mu^-,\nu}$ is defined by C.4.

2: Compute next occupancy measure for all $(x,a,x')$:

$$q_{k+1}(x,a,x') = \frac{S_{k,h}^{\mu^+,\mu^-,\nu}(x,a,x')}{\sum_{(x,a,x') \in \mathcal{X}_h \times \mathcal{A} \times \mathcal{X}_{h+1}} S_{k,h}^{\mu^+,\mu^-,\nu}(x,a,x')},$$

where $h = h(x)$ is the index of the layer of the state $x$.

3: Compute next policy for all $(x,a)$

$$\pi_{k+1}(a|x) = \frac{\sum_{x'} q_{k+1}(x,b,x')}{\sum_{b \in \mathcal{A}} \sum_{x'} q_{k+1}(x,b,x')}.$$

4: **output:** $(q_{k+1}, \pi_{k+1})$

---

## C.3 PRIVATE UOB-LBPS ALGORITHM

---

**Algorithm 3** Private UOB-LBPS

---

**Parameters:** state space $\mathcal{X}$, action space $\mathcal{A}$, episode number $K$, episode length $H$, learning rate $\eta > 0$, confidence parameter $\delta \in (0, 1]$, privacy parameter $\varepsilon > 0$ and a Privatizer

**Initialization:**

Initialize confidence set $\mathcal{P}_1$ as the set of all transitions.

For all $(x, a, x') \in \mathcal{X} \times \mathcal{A} \times \mathcal{X}_{h(x)+1}$, initialize private counts, $\widetilde{N}_1(x, a, x') = \widetilde{N}_1(x, a) = 0$, and occupancy measure, $q_1(x, a, x') = \frac{1}{X_{h(x)} A X_{h(x)+1}}$.

Initialize policy $\pi_1 = \pi^{q_1}$.

Set precision levels $E_{\varepsilon,\delta}$ and $E'_{\varepsilon,\delta}$ for Privatizer.

1: **for** $k = 1$ to $K$ **do**
2:     Execute policy $\pi_k$ for $H$ steps and obtain interaction history $\left\{ (x_h^k, a_h^k) \right\}_{h=0}^{H-1}$, and observe losses $\left\{ \ell_k(x_h^k, a_h^k) \right\}_{h=0}^{H-1}$.

3:     Receive private counts $\widetilde{N}_{k+1}(x, a)$, $\widetilde{N}_{k+1}(x, a, x')$ and private loss $\tilde{\ell}_k(x, a)$ from Privatizer.
4:     Scale private loss to $[0, 1]$, for all $(x, a) \in \mathcal{X} \times \mathcal{A}$,

$$\ddot{\ell}_k(x, a) = \frac{\tilde{\ell}_k(x, a) + E'_{\varepsilon,\delta}}{2E'_{\varepsilon,\delta} + 1}.$$

5:     Compute optimistic loss estimator $\widehat{\ell}_k(x, a)$ for all $(x, a) \in \mathcal{X} \times \mathcal{A}$ using upper occupancy measure bound $u_k$,

$$\widehat{\ell}_k(x, a) = \frac{\ddot{\ell}_k(x, a)}{u_k(x, a)},$$

    where $u_k(x, a) = \max_{P \in \mathcal{P}_k} q^{P, \pi_k}(x, a)$.

6:     Compute private transition estimate, for all $(x, a, x') \in \mathcal{X} \times \mathcal{A} \times \mathcal{X}_{h(x)+1}$,

$$\widetilde{P}_{k+1}(x'|x, a) = \frac{\widetilde{N}_{k+1}(x, a, x')}{\widetilde{N}_{k+1}(x, a)}.$$

7:     Update confidence set $\mathcal{P}_{k+1}$ using Equation (3.1),

$$\mathcal{P}_{k+1} = \left\{ P : \left| P(x'|x, a) - \widetilde{P}_{k+1}(x'|x, a) \right| \leq \beta_{k+1}(x'|x, a), \quad \forall (x, a, x') \in \mathcal{X}_h \times \mathcal{A} \times \mathcal{X}_{h+1}, h \in [H] \right\}.$$

8:     Update the occupancy measure

$$q_{k+1} = \operatorname*{argmin}_{q \in \Delta(\mathcal{P}_{k+1})} \left\langle \widehat{\ell}_k, q \right\rangle + \frac{1}{\eta} \mathcal{D}_\psi(q \| q_k).$$

    where $\psi(q)$ is a log-barrier regularizer defined in Eq. 4.4, and $\mathcal{D}_\psi(q \| q_k)$ is the Bregman divergence of $\psi$ between $q$ and $q_k$.

9:     Update the policy

$$\pi_{k+1} = \pi^{q_{k+1}}.$$

10: **end for**

---

# D OMITTED DETAILS FOR THE ANALYSIS

**Important Notations** For convenience, we define

$$\widehat{N}_k(x,a) = \max\left\{\widetilde{N}_k(x,a), 4E_{\varepsilon,\delta} + 7\ln\iota\right\}, \tag{D.1}$$

and it can be verified that the confidence width defined in Eq. 3.1 can be equivalently written as

$$\beta_k\left(x'|x,a\right) = \min\left\{1, \sqrt{\frac{2\widetilde{P}_k\left(x'|x,a\right)\left(1 - \widetilde{P}_k\left(x'|x,a\right)\right)\ln\iota}{\widetilde{N}_k(x,a)}} + \frac{4E_{\varepsilon,\delta} + 7\ln\iota}{\widetilde{N}_k(x,a)}\right\},$$

$$= \min\left\{1, \sqrt{\frac{2\widetilde{P}_k\left(x'|x,a\right)\left(1 - \widetilde{P}_k\left(x'|x,a\right)\right)\ln\iota}{\widehat{N}_k(x,a)}} + \frac{4E_{\varepsilon,\delta} + 7\ln\iota}{\widehat{N}_k(x,a)}\right\}$$

since whenever $\widehat{N}_k(x,a) \neq \widetilde{N}_k(x,a)$, the two definitions both lead to a value of 1.

## D.1 REGRET GUARANTEE IN THE FULL-INFORMATION SETTING

We decompose the regret in the same way as the non-private setting [Rosenberg and Mansour, 2019a], and bound ERROR and REG terms separately:

$$\mathcal{R}_K = \sum_{k=1}^{K}\left\langle q^{P,\pi_k} - q^*, \ell_k\right\rangle = \underbrace{\sum_{k=1}^{K}\left\langle q^{P,\pi_k} - q^{P_k,\pi_k}, \ell_k\right\rangle}_{\text{ERROR}} + \underbrace{\sum_{k=1}^{K}\left\langle q^{P_k,\pi_k} - q^*, \ell_k\right\rangle}_{\text{REG}}, \tag{D.2}$$

where $q^{P_k,\pi_k} := q_k$ is an intermediate variable that helps bound the regret, and the transition estimate $P_k$ is associated with the occupancy measure $q_k$.

### D.1.1 Bounding ERROR

**Lemma D.1** *With Assumption 2.3 and Assumption 3.2, with high probability $1 - 7\delta$, we have*

$$\text{ERROR} \leq \mathcal{O}\left(HC_M\sqrt{K} + HX^2A\ln\iota\log K + HX^2AE_{\varepsilon,\delta}\log K\right).$$

**Proof** For this proof, we consider that events in Lemma E.15 and $\mathcal{E}_{EST}$ in Proposition E.23 hold with high probability. Since for every state-action pair and episode $\ell_k(x,a) \in [0,1]$, and by Hölder's inequality, we have

$$\sum_{k=1}^{K}\left\langle q^{P,\pi_k} - q^{P_k,\pi_k}, \ell_k\right\rangle$$

$$\leq \sum_{k=1}^{K}\sum_{h=0}^{H-1}\sum_{x\in\mathcal{X}_h}\sum_{a\in\mathcal{A}}\left|q^{P_k,\pi_k}(x,a) - q^{P,\pi_k}(x,a)\right|$$

$$= \sum_{k=1}^{K}\sum_{h=0}^{H-1}\sum_{x\in\mathcal{X}_h}\left|q^{P_k,\pi_k}(x) - q^{P,\pi_k}(x)\right|$$

$$\leq \sum_{k=1}^{K}\sum_{h=0}^{H-1}\sum_{x\in\mathcal{X}_h}\sum_{h'=0}^{h-1}\sum_{(u,v,w)\in W_h'}q^{P,\pi_k}(u,v)\left|P_k(w|u,v) - P(w|u,v)\right|q^{P_k,\pi_k}(x|w)$$

$$\leq H\sum_{k=1}^{K}\sum_{h=0}^{H-1}\sum_{u\in\mathcal{X}_h}\sum_{v\in\mathcal{A}}q^{P,\pi_k}(u,v)\left\|P_k(\cdot|u,v) - P(\cdot|u,v)\right\|_1$$

$$\leq H \sum_{h=0}^{H-1} \sum_{k=1}^{K} \sum_{u \in \mathcal{X}_h} \sum_{v \in \mathcal{A}} q^{P,\pi_k}(u,v) \cdot \mathcal{O}\left(\sqrt{\frac{C_{u,v} \ln \iota}{\widehat{N}_k(u,v)}} + \frac{X_{h+1}(E_{\varepsilon,\delta} + \ln \iota)}{\widehat{N}_k(u,v)}\right)$$

$$\leq \mathcal{O}\left(H \sum_{h=0}^{H-1} \left(\sqrt{\ln \iota \sum_{(x,a) \in \mathcal{X}_h \times \mathcal{A}} C_{x,a} K} + \left(\sqrt{X_{h+1} \ln \iota} + X_{h+1}(E_{\varepsilon,\delta} + \ln \iota)\right)(X_h A \log K + \ln \iota)\right)\right)$$

$$\leq \mathcal{O}\left(H C_M \sqrt{K} + H X^2 A \ln \iota \log K + H X^2 A E_{\varepsilon,\delta} \log K\right), \tag{D.3}$$

where the third step applies Lemma E.19 and the fourth step rearranges the summation and uses the fact that $\sum_{h=0}^{H-1} \sum_{x \in \mathcal{X}_h} q^{P_k,\pi_k}(x|w) \leq H$; the fifth step follows the bound of $L_1$ norm error between transitions within the confidence set in Lemma E.16; the final steps follows the corollary of $\mathcal{E}_{EST}$ in Lemma E.24. ∎

### D.1.2 Bounding REG

To bound REG, we apply a technique similar to that utilized in the private Follow-The-Regularized-Leader (FTRL) algorithm for private online learning as described in Theorem 3.4 in Agarwal and Singh [2017] and Lemma 30 in Agarwal et al. [2023].

**Lemma D.2** *With private loss satisfying Assumption 3.2, and parameter $\eta = \sqrt{\frac{\ln(XA/H)}{K}}$ and $\delta = \frac{XA}{K}$, we have*

$$\mathbb{E}[\text{REG}] \leq 2H\sqrt{K \ln\left(\frac{XA}{H}\right)} + H E_\varepsilon. \tag{D.4}$$

**Proof** Note that $Z_k(x,a)$ is the difference between the cumulative loss of state-action pair $L_k(x,a)$ and its private version $\widetilde{L}_k(x,a)$ in $k$-th episode, i.e., $Z_k(x,a) = \widetilde{L}_k(x,a) - L_k(x,a)$. In accordance with Assumption 3.2 for private cumulative losses, we can infer that $Z_k(x,a)$ follows the same distribution and is sampled independently at each step for all $(k,x,a)$.

For ease of analysis, we introduce a pseudo-private algorithm that performs a one-shot noise injection, which follows the same distribution as $Z_k(x,a)$, but is sampled only once at the beginning of the algorithm. The learning agent then operates based on the loss with this one-shot noise. Although the pseudo-private algorithm may not be differentially private, it experiences the same expected regret as our private algorithm due to the equal expected loss.

Formally, consider a one-shot sampled noise $\widehat{Z}(x,a)$ that is independently sampled before the interaction begins. Using this one-shot noise, we define a pseudo-private cumulative loss $\widehat{L}_k(x,a) = L_k(x,a) + \widehat{Z}(x,a)$ for all $(x,a)$ pair at all episodes. Next, we establish the sequence of pseudo occupancy measures $\widehat{q}$ as follow:

$$\widehat{q}_1 \triangleq q_1, \quad \widehat{q}_k \triangleq \underset{q \in \Delta(\mathcal{P}_k)}{\operatorname{argmin}} \left\langle \widehat{L}_k, q \right\rangle + \frac{1}{\eta}\psi(q).$$

In expectation, it can be observed that $\mathbb{E}_{Z_k}[\langle q_k, \ell_k \rangle] = \mathbb{E}_{\widehat{Z}}[\langle \widehat{q}_k, \ell_k \rangle]$ since $\widehat{q}_k$ has the same distribution as $q_k$. This leads to the following equality,

$$\mathbb{E}_{Z_1,\cdots,Z_K}\left[\sum_{k=1}^{K} \langle q_k, \ell_k \rangle\right] = \mathbb{E}_Z\left[\sum_{k=1}^{K} \langle \widehat{q}_k, \ell_k \rangle\right].$$

Therefore, the pseudo-private algorithm experiences the same regret in expectation as our private algorithm, and it is sufficient to bound the regret of sequence $\{\widehat{q}_k\}_{k=1}^{K}$. The proof follows the standard template of FTRL analysis in Hazan [2016], and the key intuition is that the addition of one-shot noise does not impact the stability term of the FTRL analysis, but incurs a cost in the bias term instead.

We define an augmented series of loss function as $\ell_0(q) = \left\langle \widehat{Z}, q \right\rangle + \frac{1}{\eta}\psi(q)$. By applying the "Be the Leader" Lemma in Hazan [2016], we obtain that for any fixed $u \in \cap_k \Delta(\mathcal{P}_k)$ in the confidence set,

$$\sum_{k=0}^{K} \langle u, \ell_k \rangle \geq \sum_{k=0}^{K} \langle \widehat{q}_{k+1}, \ell_k \rangle.$$

Consequently, we can conclude that

$$\sum_{k=1}^{K} \langle \widehat{q}_k - u, \ell_k \rangle \leq \sum_{k=1}^{K} \langle \widehat{q}_k - \widehat{q}_{k+1}, \ell_k \rangle + \langle u - \widehat{q}_1, \ell_0 \rangle$$

$$\leq \sum_{k=1}^{K} \langle \widehat{q}_k - \widehat{q}_{k+1}, \ell_k \rangle + \frac{1}{\eta} R_\psi + R_{\widehat{Z}},$$

where the two bias terms are $R_\psi \triangleq \max_{q \in \Delta(P)} \psi(q) - \min_{q \in \Delta(P)} \psi(q)$, and $R_{\widehat{Z}} \triangleq \max_{q \in \Delta(P)} \langle q, \widehat{Z} \rangle - \min_{q \in \Delta(P)} \langle q, \widehat{Z} \rangle$. Via Jensen's inequality, we have $R_\psi \leq H \ln \frac{XA}{H}$. By Assumption 3.2, we have $\mathbb{E}\left[R_{\widehat{Z}}\right] \leq HE_\varepsilon$. Following Hazan [2016], the stability term is bounded by the $L_\infty$ norm of the loss function,

$$\langle \widehat{q}_k - \widehat{q}_{k+1}, \ell_k \rangle \leq \eta \|\ell_k\|_\infty^2 \leq \eta H. \tag{D.5}$$

Using Lemma 3.1, we have $q^* \in \cap_k \Delta(\mathcal{P}_k)$ with probability at least $1 - 6\delta$. Thus, putting everything together and letting $\eta = \sqrt{\frac{\ln(XA/H)}{K}}$, we obtain the following expected bound, with probability at least $1 - 6\delta$ (with probability of at most $3\delta$, we can bound it as $KH$ and setting $\delta = \frac{XA}{K}$ eliminates this term),

$$\mathbb{E}\left[\text{REG}\right] \leq 2H\sqrt{K \ln\left(\frac{XA}{H}\right)} + HE_\varepsilon.$$

$$\blacksquare$$

Putting everything together and using Lemma E.1, we obtain the expected regret bound (Theorem 3.3),

$$\mathbb{E}[\mathcal{R}_K] = \mathbb{E}\left[\text{ERROR}\right] + \mathbb{E}\left[\text{REG}\right].$$
$$\leq \mathcal{O}\left(HC_M\sqrt{K} + HX^2 AE_{\varepsilon,\delta} + HE_\varepsilon\right).$$

## D.2  REGRET GUARANTEE IN THE BANDIT SETTING

Our analysis starts from a decomposition similar to Appendix C.3 in Lee et al. [2020] as follows.

$$\mathcal{R}_K = \sum_{k=1}^{K} \langle q^{P,\pi_k} - q^*, \ell_k \rangle = \underbrace{\sum_{k=1}^{K} \langle q^{P,\pi_k} - q^{P_k,\pi_k}, \ell_k \rangle}_{\text{ERROR}} + \underbrace{\sum_{k=1}^{K} \left\langle q^{P_k,\pi_k} - \acute{q}, \ell_k - \widehat{\ell}_k \right\rangle}_{\text{BIAS}}$$

$$+ \underbrace{\sum_{k=1}^{K} \left\langle q^{P_k,\pi_k} - \acute{q}, \widehat{\ell}_k \right\rangle}_{\text{REG}} + \sum_{k=1}^{K} \langle \acute{q} - q^*, \ell_k \rangle. \tag{D.6}$$

Here, the $\acute{q}$ is defined as

$$\acute{q} = \left(1 - \frac{1}{K}\right) q^* + \frac{1}{AK} \sum_{a \in \mathcal{A}} q^{P_0,\pi_a},$$

where $\pi_a$ is the policy that chooses action $a$ at every state, and a specific transition $P_0$ is defined in Lemma C.4 in Lee et al. [2020], satisfying $P_0 \in \cap_k \mathcal{P}_k$ and $P_0(x'|, x, a) \geq \frac{1}{KX}$ for all $h < H, (x, a, x') \in \mathcal{X}_h \times \mathcal{A} \times \mathcal{X}_{h+1}$. Besides, according to Lemma C.4 in Lee et al. [2020], we also have $q^{P_0,\pi_a} \in \cap_k \Delta(\mathcal{P}_k)$, and $\acute{q}$ is also in that convex set due to convex combination rule. Note that the last term can be trivially bounded by,

$$\sum_{k=1}^{K} \langle \acute{q} - q^*, \ell_k \rangle \leq \frac{1}{AK} \sum_{k=1}^{K} \langle q^{P_0,\pi_a}, \ell_k \rangle \leq H.$$

Then, we bound each term as follows.

### D.2.1 Bounding ERROR

Since we use the same estimator for the transition function for both the full-information setting and bandit-feedback setting, the term ERROR can be bounded by using Lemma D.1. That is, with probability at least $1 - 7\delta$,

$$\text{ERROR} \leq \mathcal{O}\left(HC_M\sqrt{K} + HX^2A\ln\iota\log K + HX^2AE_{\varepsilon,\delta}\log K\right).$$

### D.2.2 Bounding BIAS

For all $k, x, a$, we define intermediate variables $g_k(x, a) = \frac{\ell_k(x,a)}{2E'_{\varepsilon,\delta}+1}$. Thus, BIAS term can be decomposed as follows,

$$\text{BIAS} = \sum_{k=1}^{K}\left\langle q^{P_k,\pi_k} - \acute{q}, \ell_k - \widehat{\ell}_k\right\rangle = \underbrace{\sum_{k=1}^{K}\left\langle q^{P_k,\pi_k} - \acute{q}, \ell_k - g_k\right\rangle}_{\text{BIAS}_1} + \underbrace{\sum_{k=1}^{K}\left\langle q^{P_k,\pi_k}, g_k - \widehat{\ell}_k\right\rangle}_{\text{BIAS}_2} + \underbrace{\sum_{k=1}^{K}\left\langle \acute{q}, \widehat{\ell}_k - g_k\right\rangle}_{\text{BIAS}_3}.$$

Before we bound the three terms, we restate the construction of our loss estimator. We first privatize the observed loss $\ell_k(x, a)\,\mathbb{I}_k(x, a)$ to $\tilde{\ell}_k(x, a)$, and then scale it to $[0, 1]$ to get the intermediate loss $\ddot{\ell}_k$. By using the upper occupancy measure $u_k$, we finally get the loss estimator $\widehat{\ell}_k(x, a)$. That is,

$$\widehat{\ell}_k(x, a) = \frac{\ddot{\ell}_k(x,a)}{u_k(x,a)} = \frac{1}{u_k(x,a)} \cdot \frac{\ell_k(x,a)\,\mathbb{I}_k(x,a) + Z_k(x,a) + E'_{\varepsilon,\delta}}{2E'_{\varepsilon,\delta}+1}. \tag{D.7}$$

Observe that the estimated loss $\widehat{\ell}_k$ is biased when given previous trajectories $S_{1:k-1}$, since $P_k$ may be different from the true transition function $P$ and the losses are perturbed by noise and then scaled. The randomness comes from the random policy, stochastic transition function, and zero-mean injected noise.

$$\mathbb{E}\left[\widehat{\ell}_k(x, a)\,|S_{1:k-1}\right] = \frac{1}{u_k(x,a)} \cdot \frac{E'_{\varepsilon,\delta}}{2E'_{\varepsilon,\delta}+1} + \frac{q^{P,\pi_k}(x,a)}{u_k(x,a)} \cdot \frac{\ell_k(x,a)}{2E'_{\varepsilon,\delta}+1}. \tag{D.8}$$

**BIAS$_1$:** By definition and nice decomposition, we have BIAS$_1$ controlled by $\mathcal{R}_K - \text{ERROR}$.

$$\begin{aligned}
\text{BIAS}_1 &= \sum_{k=1}^{K}\left\langle q^{P_k,\pi_k} - \acute{q}, \ell_k - g_k\right\rangle = \frac{2E'_{\varepsilon,\delta}}{2E'_{\varepsilon,\delta}+1}\sum_{k=1}^{K}\left\langle q^{P_k,\pi_k} - \acute{q}, \ell_k\right\rangle \\
&\leq \frac{2E'_{\varepsilon,\delta}}{2E'_{\varepsilon,\delta}+1}\sum_{k=1}^{K}\left\langle q^{P_k,\pi_k} - q^*, \ell_k\right\rangle \\
&= \frac{2E'_{\varepsilon,\delta}}{2E'_{\varepsilon,\delta}+1}\left[\sum_{k=1}^{K}\left\langle q^{P,\pi_k} - q^*, \ell_k\right\rangle - \sum_{k=1}^{K}\left\langle q^{P,\pi_k} - q^{P_k,\pi_k}, \ell_k\right\rangle\right] \\
&= \frac{2E'_{\varepsilon,\delta}}{2E'_{\varepsilon,\delta}+1}\left[\mathcal{R}_K - \text{ERROR}\right].
\end{aligned}$$

**BIAS$_2$:** In expectation, we have,

$$\begin{aligned}
\mathbb{E}\left[\text{BIAS}_2\right] &= \mathbb{E}\left[\sum_{k=1}^{K}\left\langle q^{P_k,\pi_k}, g_k - \widehat{\ell}_k\right\rangle\right] \\
&= \mathbb{E}\left[\sum_{k=1}^{K}\sum_{x,a}q^{P_k,\pi_k}(x,a)\left(\frac{\ell_k(x,a)}{2E'_{\varepsilon,\delta}+1} - \frac{1}{u_k(x,a)}\cdot\frac{E'_{\varepsilon,\delta}}{2E'_{\varepsilon,\delta}+1} - \frac{q^{P,\pi_k}(x,a)}{u_k(x,a)}\cdot\frac{\ell_k(x,a)}{2E'_{\varepsilon,\delta}+1}\right)\right] \\
&= \mathbb{E}\left[\sum_{k=1}^{K}\sum_{x,a}q^{P_k,\pi_k}(x,a)\left(\frac{u_k(x,a) - q^{P,\pi_k}(x,a)}{u_k(x,a)}\cdot\frac{\ell_k(x,a)}{2E'_{\varepsilon,\delta}+1} - \frac{1}{u_k(x,a)}\cdot\frac{E'_{\varepsilon,\delta}}{2E'_{\varepsilon,\delta}+1}\right)\right] \\
&\leq \mathbb{E}\left[\frac{1}{2E'_{\varepsilon,\delta}+1}\cdot\sum_{k=1}^{K}\sum_{x}\left|u_k(x) - q^{P,\pi_k}(x)\right| - \sum_{k=1}^{K}\sum_{x,a}\frac{1}{u_k(x,a)}\cdot\frac{E'_{\varepsilon,\delta}}{2E'_{\varepsilon,\delta}+1}\right],
\end{aligned}$$

where the second equality applies the expectation of private loss estimator in Eq. D.8; and the fourth inequality is due to the definition of the upper occupancy measure, i.e., $u_k(x,a) \geq q^{P_k,\pi_k}$; Lemma 3.1, i.e., $P \in \mathcal{P}_k$ with high probability, and $\ell_k(x,a) \in [0,1]$ for all $(x,a,k)$.

Similar to the proof of Lemma D.1, we consider that the event in Lemma E.15 and $\mathcal{E}_{EST}$ in Proposition E.23 hold with high probability. Then, we have

$$
\sum_{x \in \mathcal{X}} \left| u_k(x) - q^{P,\pi_k}(x) \right|
$$

$$
\leq \mathcal{O}\left( \sum_{x \in \mathcal{X}} \sum_{h=0}^{h(x)-1} \sum_{(u,v,w) \in W_h} q^{P,\pi_k}(u,v) \cdot \sqrt{\frac{P(w|u,v)(1-P(w|u,v))\ln \iota}{\widehat{N}_k(u,v)}} \cdot q^{P,\pi_k}(x|w) \right)
$$

$$
+ \mathcal{O}\left( X^3 \sum_{u \neq x_H} \sum_{v \in \mathcal{A}} q^{P,\pi_k}(u,v) \cdot \frac{E_{\varepsilon,\delta} + \ln \iota}{\widehat{N}_k(u,v)} \right)
$$

$$
\leq \mathcal{O}\left( H \sum_{h=0}^{h(x)-1} \sum_{(u,v,w) \in W_h} q^{P,\pi_k}(u,v) \cdot \sqrt{\frac{P(w|u,v)(1-P(w|u,v))\ln \iota}{\widehat{N}_k(u,v)}} \right)
$$

$$
+ \mathcal{O}\left( X^3 \sum_{u \neq x_H} \sum_{v \in \mathcal{A}} q^{P,\pi_k}(u,v) \cdot \frac{E_{\varepsilon,\delta} + \ln \iota}{\widehat{N}_k(u,v)} \right)
$$

$$
\leq \mathcal{O}\left( H \sum_{h=0}^{H-1} \sum_{u \in x_h} \sum_{v \in \mathcal{A}} q^{P,\pi_k}(u,v) \cdot \sqrt{\frac{C_{u,v} \ln \iota}{\widehat{N}_k(u,v)}} \right) + \mathcal{O}\left( X^3 \sum_{u \neq x_H} \sum_{v \in \mathcal{A}} q^{P,\pi_k}(u,v) \cdot \frac{E_{\varepsilon,\delta} + \ln \iota}{\widehat{N}_k(u,v)} \right),
$$

where the first step applies the occupancy difference lemma in Lemma E.21, and the second inequality is due to the fact that $\sum_{x \in \mathcal{X}} q^{P,\pi_k}(x|w) \leq H$.

Taking the summation over all episodes yields the following:

$$
\sum_{k=1}^{K} \sum_{x \in \mathcal{X}} \left| u_k(x) - q^{P,\pi_k}(x) \right|
$$

$$
\leq \mathcal{O}\left( H \sum_{k=1}^{K} \sum_{h=0}^{H-1} \sum_{u \in x_h} \sum_{v \in \mathcal{A}} q^{P,\pi_k}(u,v) \cdot \sqrt{\frac{C_{u,v} \ln \iota}{\widehat{N}_k(u,v)}} \right) + \mathcal{O}\left( X^3 \sum_{k=1}^{K} \sum_{u \neq x_H} \sum_{v \in \mathcal{A}} q^{P,\pi_k}(u,v) \cdot \frac{E_{\varepsilon,\delta} + \ln \iota}{\widehat{N}_k(u,v)} \right)
$$

$$
\leq \mathcal{O}\left( H \sum_{h=0}^{H-1} \sqrt{\ln \iota} \left( \sqrt{\sum_{(x,a) \in \mathcal{X}_h \times \mathcal{A}} C_{x,a} K} + \sqrt{X_{h+1}} X_h A \log K + \sqrt{X_{h+1}} \log \iota \right) \right)
$$

$$
+ \mathcal{O}\left( X^3 \sum_{h=0}^{H-1} (E_{\varepsilon,\delta} + \ln \iota) (X_h A \log K + \log \iota) \right)
$$

$$
\leq \mathcal{O}\left( H C_M \sqrt{K \ln \iota} + X^4 A (E_{\varepsilon,\delta} + \ln \iota) \ln \iota \right),
$$

where the second inequality follows the corollary of $\mathcal{E}_{EST}$ in Lemma E.24. Thus, we derive the upper bound of $\text{BIAS}_2$,

$$
\mathbb{E}[\text{BIAS}_2] \leq \mathcal{O}\left( \frac{1}{2E'_{\varepsilon,\delta} + 1} \cdot \left( H C_M \sqrt{K} + X^4 A E_{\varepsilon,\delta} \right) \right) - \mathbb{E}\left[ \sum_{k=1}^{K} \sum_{x,a} \frac{1}{u_k(x,a)} \cdot \frac{E'_{\varepsilon,\delta}}{2E'_{\varepsilon,\delta} + 1} \right].
$$

**BIAS$_3$:** We apply the Eq.D.8 and the definition of upper occupancy measure $u_k(x,a)$ and Lemma 3.1 directly, then obtain

BIAS$_3$ bounded.

$$\mathbb{E}\left[\text{BIAS}_3\right] = \mathbb{E}\left[\sum_{k=1}^{K}\left\langle \acute{q}, \widehat{\ell}_k - g_k\right\rangle\right]$$

$$= \mathbb{E}\left[\sum_{k=1}^{K}\sum_{x,a}\acute{q}(x,a)\cdot\left(\frac{1}{u_k(x,a)}\cdot\frac{E'_{\varepsilon,\delta}}{2E'_{\varepsilon,\delta}+1} + \frac{q^{P,\pi_k}(x,a)}{u_k(x,a)}\cdot\frac{\ell_k(x,a)}{2E'_{\varepsilon,\delta}+1} - \frac{\ell_k(x,a)}{2E'_{\varepsilon,\delta}+1}\right)\right]$$

$$= \mathbb{E}\left[\sum_{k=1}^{K}\sum_{x,a}\acute{q}(x,a)\cdot\left(\frac{q^{P,\pi_k}(x,a)-u_k(x,a)}{u_k(x,a)}\cdot\frac{\ell_k(x,a)}{2E'_{\varepsilon,\delta}+1} + \frac{1}{u_k(x,a)}\cdot\frac{E'_{\varepsilon,\delta}}{2E'_{\varepsilon,\delta}+1}\right)\right]$$

$$\leq \mathbb{E}\left[\sum_{k=1}^{K}\sum_{x,a}\frac{1}{u_k(x,a)}\cdot\frac{E'_{\varepsilon,\delta}}{2E'_{\varepsilon,\delta}+1}\right].$$

### D.2.3 Bounding REG

**Lemma D.3** *With private loss estimator satisfying Assumption 4.1, and let $\eta = \sqrt{\frac{X}{K}}, \delta = \frac{XA}{K}$, we have,*

$$\mathbb{E}\left[\text{REG}\right] \leq \mathcal{O}\left(\frac{\sqrt{XK}}{2E'_{\varepsilon,\delta}+1}\cdot\left(AXE'_{\varepsilon,\delta}+H\right)\right)$$

**Proof** Using the standard analysis of OMD with log-barrier (e.g., Lemma 12 in Agarwal et al. [2017]), we have, for any $u \in \cap_k \Delta(P_k)$,

$$\sum_{k=1}^{K}\left\langle q^{P_k,\pi_k} - u, \widehat{\ell}_k\right\rangle \leq \frac{\mathcal{D}_\psi(u\|q_1)}{\eta} + \eta\sum_{k=1}^{K}\sum_{x,a,x'}q^{P_k,\pi_k}(x,a,x')^2\,\widehat{\ell}_k(x,a)^2$$

$$= \frac{\mathcal{D}_\psi(u\|q_1)}{\eta} + \eta\sum_{k=1}^{K}\sum_{x,a,x'}\left(q^{P_k,\pi_k}(x,a)\,P_k(x'|x,a)\right)^2\widehat{\ell}_k(x,a)^2$$

$$\leq \frac{\mathcal{D}_\psi(u\|q_1)}{\eta} + \eta\sum_{k=1}^{K}\sum_{x,a}q^{P_k,\pi_k}(x,a)^2\,\widehat{\ell}_k(x,a)^2\cdot\left(\sum_{x'}P_k(x'|x,a)^2\right)$$
(D.9)

$$\leq \frac{\mathcal{D}_\psi(u\|q_1)}{\eta} + \eta\sum_{k=1}^{K}\sum_{x,a}q^{P_k,\pi_k}(x,a)^2\,\widehat{\ell}_k(x,a)^2,$$

where the first step is due to $\eta q^{P_k,\pi_k}(x,a,x')\,\widehat{\ell}_k(x,a) \geq 0$ under the Assumption 4.1, and the second step uses the fact that $q^{P_k,\pi_k}(x,a,x') = q^{P_k,\pi_k}(x,a)\,P_k(x'|x,a)$; the final step follows the fact that $\sum_{x'}P_k(x'|x,a) = 1$.

Eq. D.9 also applies for $\acute{q}$, since $\acute{q} \in \cap_k\Delta(P_k)$ by definition. Therefore, by direct calculation, we have

$$\mathcal{D}_\psi(\acute{q}\|q_1) = \sum_{h=0}^{H-1}\sum_{(x,a,x')\in W_h}\left(\log\left(\frac{q_1(x,a,x')}{\acute{q}(x,a,x')}\right) + \frac{\acute{q}(x,a,x')}{q_1(x,a,x')} - 1\right)$$

$$= \sum_{h=0}^{H-1}\sum_{(x,a,x')\in W_h}\log\left(\frac{q_1(x,a,x')}{\acute{q}(x,a,x')}\right) + \sum_{h=0}^{H-1}\sum_{(x,a,x')\in W_h}\left(X_h A X_{h+1}\acute{q}(x,a,x') - 1\right)$$

$$= \sum_{h=0}^{H-1}\sum_{(x,a,x')\in W_h}\log\left(\frac{q_1(x,a,x')}{\acute{q}(x,a,x')}\right)$$

$$\leq 3X^2 A\log\iota,$$

where the second step uses the definition of $q_1(x,a,x') = \frac{1}{X_h A X_{h+1}}$ for all horizon $h$, and the fourth step uses the lower bounds $\acute{q}(x,a,x') \geq \frac{1}{X^2 A K^3}$ from Lemma C.10 in Lee et al. [2020], and then $\acute{q}(x,a,x') \geq \frac{1}{\iota^3}$ and upper bounds

$q_1(x, a, x') \leq 1$. For the second term, we have

$$\mathbb{E}\left[\sum_{k=1}^{K}\sum_{x,a} q^{P_k,\pi_k}(x,a)^2\, \widehat{\ell}_k(x,a)^2\right]$$

$$= \mathbb{E}\left[\sum_{k=1}^{K}\sum_{x,a} q^{P_k,\pi_k}(x,a)^2 \cdot \frac{\ddot{\ell}_k(x,a)}{u_k(x,a)} \cdot \widehat{\ell}_k(x,a)\right]$$

$$\leq \mathbb{E}\left[\sum_{k=1}^{K}\sum_{x,a} q^{P_k,\pi_k}(x,a) \cdot \widehat{\ell}_k(x,a)\right]$$

$$\leq \mathbb{E}\left[\sum_{k=1}^{K}\sum_{x,a} q^{P_k,\pi_k}(x,a) \cdot \left(\frac{1}{u_k(x,a)} \cdot \frac{E'_{\varepsilon,\delta}}{2E'_{\varepsilon,\delta}+1} + \frac{q^{P,\pi_k}(x,a)}{u_k(x,a)} \cdot \frac{\ell_k(x,a)}{2E'_{\varepsilon,\delta}+1}\right)\right]$$

$$\leq \mathbb{E}\left[\sum_{k=1}^{K}\sum_{x,a} \left(\frac{E'_{\varepsilon,\delta}}{2E'_{\varepsilon,\delta}+1} + q^{P,\pi_k}(x,a) \cdot \frac{\ell_k(x,a)}{2E'_{\varepsilon,\delta}+1}\right)\right]$$

$$\leq \frac{XAKE'_{\varepsilon,\delta} + HK}{2E'_{\varepsilon,\delta}+1},$$

where the first equality applies the definition of $\widehat{\ell}_k(x,a)$, and the second inequality follows the definition of $u_k$ and the fact that $\ddot{\ell}_k(x,a) \in [0,1]$ for all $(x,a)$ due to the scaling procedure; the third inequality follows the expectation of $\widehat{\ell}_k(x,a)$ in Eq.D.8, and the fourth inequality applies the definition of $u_k$ again, and the fact that $\ell_k \in [0,1]$.

Thus, setting $\eta = \sqrt{\frac{X}{K}}, \delta = \frac{XA}{K}$ we obtain the bound of REG,

$$\mathbb{E}\left[\sum_{k=1}^{K}\left\langle q^{P_k,\pi_k} - \acute{q}, \widehat{\ell}_k\right\rangle\right] \leq \mathcal{O}\left(\frac{\sqrt{XK}}{2E'_{\varepsilon,\delta}+1} \cdot \left(AXE'_{\varepsilon,\delta} + H\right)\right) \tag{D.10}$$

∎

Putting everything together and using Lemma E.1, we obtain the expected regret bound (Theorem 4.2),

$$\mathbb{E}[\mathcal{R}_K] = \mathbb{E}\left[\text{ERROR}\right] + \left(2E'_{\varepsilon,\delta}+1\right) \cdot \left(\mathbb{E}\left[\text{BIAS}_2\right] + \mathbb{E}\left[\text{REG}\right]\right).$$

$$\leq \widetilde{\mathcal{O}}\left(HC_M\sqrt{K} + HX^4AE_{\varepsilon,\delta} + AE'_{\varepsilon,\delta}\sqrt{X^3K}\right).$$

# E    SUPPLEMENTARY LEMMAS

## E.1    USEFUL RESULTS

**Lemma E.1 (Expectation, Lemma D.3.6 in Jin et al. [2021])** *Suppose that a random variable $Z$ satisfies the following conditions:*

- *$Z < R$ where $R$ is a constant.*
- *$Z < Y$ conditioning on event $E$, where $Y \geq 0$ is a random variable.*

*Then, it holds that $\mathbb{E}[Z] \leq \mathbb{E}[Y] + \mathbb{P}[E^c] \cdot R$ where $E^c$ is the complementary event of $E$.*

**Lemma E.2 (Concentration of Laplace variable)** *Suppose random variable $Z \sim Lap(\lambda)$, then for any $t > 0$ we have,*

$$\mathbb{P}\left[|Z| \geq \lambda \ln t\right] \leq \frac{1}{t}.$$

**Definition E.3 (Sub-Exponential variables)** *A random variable $X$ with mean $\mu = \mathbb{E}[X]$ is sub-exponential if there are non-negative parameters $(\nu, b)$ such that*

$$\mathbb{E}\left[e^{\lambda(X-\mu)}\right] \leq e^{\frac{\lambda^2 \nu^2}{2}} \quad \forall |\lambda| < \frac{1}{b}.$$

**Lemma E.4 (Laplace variable is sub-exponential)** *A Laplace variable with parameter $(\mu, \beta)$, i.e., its probability density function is $f(x) = \frac{1}{2\beta} \exp\left(-\frac{|x-\mu|}{\beta}\right)$, is also a sub-exponential variable with parameter $(2\beta, \beta)$.*

**Lemma E.5 (Linear combination rule for sub-exponential variables)** *Consider an independent sequence of $\{X_k\}_{k=1}^n$ of random variables, such that $X_k$ has mean $\mu_k$, and is sub-exponential with parameter $(\nu_k, b_k)$. Then the variable $Z = \sum_{k=1}^n X_k$ is sub-exponential with parameters $(\nu_*, b_*)$, where*

$$\nu_* := \sqrt{\sum_{k=1}^n v_k^2} \qquad b_* := \max_{k=1,\dots,n} b_k.$$

**Lemma E.6 (Maxima of sub-exponential variables)** *Let $\boldsymbol{Z} \triangleq (Z_1, \cdots, Z_n)$, and for all $i \in [n]$, $Z_i$ is a independent identical distributed sub-exponential random variable with parameter $(\nu, b)$. Then, if $b\sqrt{2\ln n} < \nu$, we have*

$$\mathbb{E}\left[\max_{i \in [n]} Z_i\right] \leq \nu\sqrt{2\ln n}.$$

**Proof** With the definition of sub-exponential variables in Def. E.3, the following inequality holds for all $\lambda$ with $|\lambda| < \frac{1}{b}$, by using Jensen's inequality,

$$\exp\left(\lambda \mathbb{E} \max_{i \in [n]} Z_i\right) \leq \mathbb{E} \exp\left(\lambda \max_{i \in [n]} Z_i\right) = \mathbb{E} \max_{i \in [n]} e^{\lambda Z_i} \leq \sum_{i \in [n]} \mathbb{E} e^{\lambda Z_i} \leq n e^{\frac{\lambda^2 \nu^2}{2}}.$$

Taking logarithms on both sides, we have $\mathbb{E} \max_{i \in [n]} Z_i \leq \frac{\ln n}{\lambda} + \frac{\lambda \nu^2}{2}$. Therefore, the upper bound is minimized when $\lambda = \frac{\sqrt{2\ln n}}{\nu} \leq \frac{1}{b}$, which yields

$$\mathbb{E}\left[\max_{i \in [n]} Z_i\right] \leq \nu\sqrt{2\ln n}.$$

∎

**Lemma E.7 (Maxima of the sum of i.i.d. Laplace variables)** *Let $\boldsymbol{Z} \triangleq (Z_1, \cdots, Z_n)$, and $Z_i = \sum_j^m b_{ij}$ for all $i \in [n]$, where for all $i \in [n], j \in [m]$, $b_{ij}$ is independent identical distributed Laplace variable $\mathrm{Lap}(0, \beta)$ with parameter $(0, \beta)$. Then, if $m > \frac{\ln n}{2}$, we have*

$$\mathbb{E}\left[\max_{i \in [n]} Z_i\right] = \mathbb{E}\left[\max_{i \in [n]} \sum_{j \in [m]} b_{ij}\right] \leq 2\beta\sqrt{2m\ln n}. \tag{E.1}$$

**Proof** Since Laplace variable is also sub-exponential (refer to Lemma E.4), the sum of i.i.d. Laplace variable is also a sub-exponential variable with parameter $(2\beta\sqrt{m}, \beta)$ according to the linear combination rule of sub-exponential variables in Lemma E.5. Then, we can obtain the lemma by using the maxima of sub-exponential variables in Lemma E.7. ∎

**Lemma E.8 (Revised Lemma 11 in Bourel et al. [2020])** *Consider $x$ and $y$ satisfying $|x - y| \leq \alpha\sqrt{y(1-y)} + \beta$. Then*

$$\sqrt{y(1-y)} \leq \sqrt{x(1-x)} + 1.9\alpha + 1.5\sqrt{\beta}.$$

**Proof** By Taylor's expansion, we have

$$
\begin{aligned}
y(1-y) &= x(1-x) + (1-2x)(y-x) - (y-x)^2 \\
&= x(1-x) + (1-x-y)(y-x) \\
&\leq x(1-x) + |1-x-y| \left( \alpha \sqrt{y(1-y)} + \beta \right) \\
&\leq x(1-x) + \alpha \sqrt{y(1-y)} + \beta
\end{aligned}
$$

Using the fact that $a \leq b\sqrt{a} + c$ implies $a \leq b^2 + b\sqrt{c} + c$ for non-negative numbers $a, b$ and $c$, we get

$$
\begin{aligned}
y(1-y) &\leq \alpha^2 + \alpha\sqrt{x(1-x) + \beta} + x(1-x) + \beta \\
&\leq \alpha^2 + \alpha\sqrt{x(1-x)} + \alpha\sqrt{\beta} + x(1-x) + \beta \\
&= \left( \sqrt{x(1-x)} + \frac{\alpha}{2} \right)^2 + \frac{3\alpha^2}{4} + \alpha\sqrt{\beta} + \beta,
\end{aligned}
$$

where we use $\sqrt{a+b} \leq \sqrt{a} + \sqrt{b}$ for all $a, b \geq 0$. Taking square-root from both sides, and applying $\sqrt{ab} \leq \frac{a+b}{2}$, we have the desired result:

$$
\sqrt{y(1-y)} \leq \sqrt{x(1-x)} + \frac{\alpha}{2} + \frac{\sqrt{3}\alpha}{2} + \frac{\alpha}{2} + \frac{\sqrt{\beta}}{2} + \sqrt{\beta} \leq \sqrt{x(1-x)} + 1.9\alpha + 1.5\sqrt{\beta}.
$$

∎

**Lemma E.9 (Lemma 19 in Jaksch et al. [2010], Lemma 24 in Talebi and Maillard [2018])** *For any sequence of numbers* $z_1, z_2, \ldots, z_n$ *with* $0 \leq z_k \leq Z_{k-1} := \max\left\{ 1, \sum_{i=1}^{k-1} z_i \right\}$, *it holds*

$$
(1) \quad \sum_{k=1}^{n} \frac{z_k}{\sqrt{Z_{k-1}}} \leq \left( \sqrt{2} + 1 \right) \sqrt{Z_n},
$$

$$
(2) \quad \sum_{k=1}^{n} \frac{z_k}{Z_{k-1}} \leq 2\log\left( Z_n \right) + 1.
$$

The next one is a standard Bernstein-type concentration inequality for martingale (Theorem 1 in Beygelzimer et al. [2011]).

**Lemma E.10** *Let* $Y_1, \ldots, Y_K$ *be a martingale difference sequence with respect to a filtration* $\mathcal{F}_1, \ldots, \mathcal{F}_K$. *Assume* $Y_k \leq R$ *a.s. for all* $k$. *Then for any* $\delta \in (0,1)$ *and* $\lambda \in [0, 1/R]$, *with probability at least* $1 - \delta$, *we have*

$$
\sum_{k=1}^{K} Y_k \leq \lambda \sum_{k=1}^{K} \mathbb{E}_k\left[Y_k^2\right] + \frac{\ln(1/\delta)}{\lambda}.
$$

### E.2 CONFIDENCE BOUND WITH PRIVACY

We first define the *non-private* empirical transition probability and the *private* empirical transition probability as follows,

$$
\bar{P}_k\left(x'|x,a\right) := \frac{N_k(x,a,x')}{N_k(x,a)}, \quad \widetilde{P}_k\left(x'|x,a\right) := \frac{\widetilde{N}_k(x,a,x')}{\widetilde{N}_k(x,a)}.
$$

**Lemma E.11 (Lemma 2, [Jin et al., 2020a])** *With probability at least* $1 - 4\delta$, *we have a good event*

$$
\forall (x,a,x') \in W_h, \forall h, k, \quad \left| P\left(x'|x,a\right) - \bar{P}_k\left(x'|x,a\right) \right| \leq \bar{\beta}_k\left(x'|x,a\right),
$$

*and* $\bar{\beta}_k\left(x'|x,a\right)$ *for any* $(x,a,x') \in W_h$ *and* $h \in [H]$ *is defined as*

$$
\bar{\beta}_k\left(x'|x,a\right) = \min\left\{ 1, \sqrt{\frac{2\bar{P}_k\left(x'|x,a\right)\left(1 - \bar{P}_k\left(x'|x,a\right)\right)\ln \iota}{N_k(x,a)}} + \frac{14\ln \iota}{3N_k(x,a)} \right\}.
$$

Then, we have the difference lemmas between the private transition estimate and the non-private transition estimate as follows.

**Lemma E.12** *For all* $(x, a, x') \in W_h, h \in [H], k \in [K]$, *we have*

$$\left| \widetilde{P}_k \left( x'|x, a \right) - \bar{P}_k \left( x'|x, a \right) \right| \leq \frac{2E_{\varepsilon,\delta}}{\widetilde{N}_k(x, a)}.$$

**Proof** By definition, we have

$$
\begin{aligned}
\left| \widetilde{P}_k \left( x'|x, a \right) - \bar{P}_k \left( x'|x, a \right) \right| &\leq \left| \frac{\widetilde{N}_k(x, a, x')}{\widetilde{N}_k(x, a)} - \frac{N_k(x, a, x')}{\widetilde{N}_k(x, a)} \right| + \left| \frac{N_k(x, a, x')}{\widetilde{N}_k(x, a)} - \frac{N_k(x, a, x')}{N_k(x, a)} \right| \\
&\leq \frac{E_{\varepsilon,\delta}}{\widetilde{N}_k(x, a)} + \frac{N_k(x, a, x')E_{\varepsilon,\delta}}{N_k(x, a) \cdot \widetilde{N}_k(x, a)} \\
&\leq \frac{2E_{\varepsilon,\delta}}{\widetilde{N}_k(x, a)}.
\end{aligned}
$$

∎

**Lemma E.13** *For all* $(x, a, x') \in W_h, h \in [H], k \in [K]$, *we have,*

$$\left| \widetilde{P}_k \left( x'|x, a \right) \left( 1 - \widetilde{P}_k \left( x'|x, a \right) \right) - \bar{P}_k \left( x'|x, a \right) \left( 1 - \bar{P}_k \left( x'|x, a \right) \right) \right| \leq \frac{4E_{\varepsilon,\delta}}{\widetilde{N}_k(x, a)}.$$

**Proof** Similar to Lemma E.12, we have,

$$
\begin{aligned}
&\left| \widetilde{P}_k \left( x'|x, a \right) \left( 1 - \widetilde{P}_k \left( x'|x, a \right) \right) - \bar{P}_k \left( x'|x, a \right) \left( 1 - \bar{P}_k \left( x'|x, a \right) \right) \right| \\
&= \left| \frac{N_k(x, a, x')}{N_k(x, a)} \cdot \frac{N_k(x, a) - N_k(x, a, x')}{N_k(x, a)} - \frac{\widetilde{N}_k(x, a, x')}{\widetilde{N}_k(x, a)} \cdot \frac{\widetilde{N}_k(x, a) - \widetilde{N}_k(x, a, x')}{\widetilde{N}_k(x, a)} \right| \\
&\leq \left| \frac{N_k(x, a) - N_k(x, a, x')}{N_k(x, a)} \left[ \frac{N_k(x, a, x')}{N_k(x, a)} - \frac{N_k(x, a, x')}{\widetilde{N}_k(x, a)} \right] \right| \\
&\quad + \left| \frac{\widetilde{N}_k(x, a) - \widetilde{N}_k(x, a, x')}{\widetilde{N}_k(x, a)} \left[ \frac{\widetilde{N}_k(x, a, x')}{\widetilde{N}_k(x, a)} - \frac{N_k(x, a, x')}{\widetilde{N}_k(x, a)} \right] \right| \\
&\quad + \left| \frac{N_k(x, a, x')}{\widetilde{N}_k(x, a)} \left[ \frac{\widetilde{N}_k(x, a, x')}{\widetilde{N}_k(x, a)} - \frac{N_k(x, a, x')}{N_k(x, a)} \right] \right| \\
&\leq \frac{N_k(x, a, x')E_{\varepsilon,\delta}}{N_k(x, a) \cdot \widetilde{N}_k(x, a)} + \frac{E_{\varepsilon,\delta}}{\widetilde{N}_k(x, a)} + \frac{N_k(x, a, x')}{N_k(x, a)} \cdot \frac{2E_{\varepsilon,\delta}}{\widetilde{N}_k(x, a)} \\
&\leq \frac{4E_{\varepsilon,\delta}}{\widetilde{N}_k(x, a)}.
\end{aligned}
$$

∎

With the help of the lemmas above, it's ready to prove Lemma 3.1 now.

**Lemma E.14 (Restatement of Lemma 3.1)** *With probability at least* $1 - 6\delta$, *we have a good event,*

$$\forall (x, a, x') \in W_h, \forall h, k, \quad \left| P \left( x'|x, a \right) - \widetilde{P}_k \left( x'|x, a \right) \right| \leq \beta_k \left( x'|x, a \right),$$

and $\beta_k \left(x'|x,a\right)$ *for any* $(x,a,x') \in W_h$ *and* $h \in [H]$ *is defined as*

$$\beta_k \left(x'|x,a\right) = \min \left\{1, \sqrt{\frac{2\widetilde{P}_k \left(x'|x,a\right) \left(1 - \widetilde{P}_k \left(x'|x,a\right)\right) \ln \iota}{\widetilde{N}_k(x,a)}} + \frac{4E_{\varepsilon,\delta} + 7\ln \iota}{\widetilde{N}_k(x,a)}\right\}$$

$$= \min \left\{1, \sqrt{\frac{2\widetilde{P}_k \left(x'|x,a\right) \left(1 - \widetilde{P}_k \left(x'|x,a\right)\right) \ln \iota}{\widehat{N}_k(x,a)}} + \frac{4E_{\varepsilon,\delta} + 7\ln \iota}{\widehat{N}_k(x,a)}\right\}.$$

**Proof** [Proof of Lemma E.14] With standard decomposition, we have,

$$\left|\widetilde{P}_k \left(x'|x,a\right) - P\left(x'|x,a\right)\right| \le \left|\frac{\widetilde{N}_k(x,a,x') - N_k(x,a,x')}{\widetilde{N}_k(x,a)}\right| + \left|\frac{N_k(x,a,x')}{\widetilde{N}_k(x,a)} - P\left(x'|x,a\right)\right|$$

$$\le \frac{E_{\varepsilon,\delta}}{\widetilde{N}_k(x,a)} + \left|\frac{N_k(x,a,x')}{N_k(x,a)} \cdot \frac{N_k(x,a)}{\widetilde{N}_k(x,a)} - P\left(x'|x,a\right)\right|$$

$$\le \frac{E_{\varepsilon,\delta}}{\widetilde{N}_k(x,a)} + \left|\frac{N_k(x,a)}{\widetilde{N}_k(x,a)} \cdot \left(\frac{N_k(x,a,x')}{N_k(x,a)} - P\left(x'|x,a\right)\right)\right| + \left|P\left(x'|x,a\right) \left(\frac{N_k(x,a)}{\widetilde{N}_k(x,a)} - 1\right)\right|$$

$$\le \frac{2E_{\varepsilon,\delta}}{\widetilde{N}_k(x,a)} + \frac{N_k(x,a)}{\widetilde{N}_k(x,a)} \cdot \left|\bar{P}_k \left(x'|x,a\right) - P\left(x'|x,a\right)\right|$$

$$\le \frac{2E_{\varepsilon,\delta}}{\widetilde{N}_k(x,a)} + \frac{N_k(x,a)}{\widetilde{N}_k(x,a)} \cdot \left(\sqrt{\frac{2\bar{P}_k \left(x'|x,a\right) \left(1 - \bar{P}_k \left(x'|x,a\right)\right) \ln \iota}{N_k(x,a)}} + \frac{14\ln \iota}{3N_k(x,a)}\right)$$

$$\le \frac{2E_{\varepsilon,\delta}}{\widetilde{N}_k(x,a)} + \sqrt{\frac{2\left(\widetilde{P}_k \left(x'|x,a\right) \left(1 - \widetilde{P}_k \left(x'|x,a\right)\right) + \frac{4E_{\varepsilon,\delta}}{\widetilde{N}_k(x,a)}\right) \ln \iota}{\widetilde{N}_k(x,a)}} + \frac{14\ln \iota}{3\widetilde{N}_k(x,a)}$$

$$\le \sqrt{\frac{2\widetilde{P}_k \left(x'|x,a\right) \left(1 - \widetilde{P}_k \left(x'|x,a\right)\right) \ln \iota}{\widetilde{N}_k(x,a)}} + \frac{2E_{\varepsilon,\delta} + 5\ln \iota + 4\sqrt{E_{\varepsilon,\delta} \ln \iota}}{\widetilde{N}_k(x,a)}$$

$$\le \sqrt{\frac{2\widetilde{P}_k \left(x'|x,a\right) \left(1 - \widetilde{P}_k \left(x'|x,a\right)\right) \ln \iota}{\widetilde{N}_k(x,a)}} + \frac{4E_{\varepsilon,\delta} + 7\ln \iota}{\widetilde{N}_k(x,a)},$$

where the fifth inequality follows Lemma E.11, and the sixth inequality follows Lemma E.13. The seventh and eighth step use $\sqrt{x+y} \le \sqrt{x} + \sqrt{y}$ and $2\sqrt{xy} \le x + y$ for $x, y \ge 0$. The second equality in the lemma follows the definition of $\widehat{N}_k(x,a)$ in the Eq.D.1. ∎

**Lemma E.15** *Conditioning on event in Lemma E.14, it holds for any episode $k$ and any transition $P' \in \mathcal{P}_k$,*

$$\left|P' \left(x'|x,a\right) - P\left(x'|x,a\right)\right| \le \min \left\{1, \sqrt{\frac{2P\left(x'|x,a\right) \left(1 - P\left(x'|x,a\right)\right) \ln \iota}{\widehat{N}_k(x,a)}} + \frac{7E_{\varepsilon,\delta} + 26\ln \iota}{\widehat{N}_k(x,a)}\right\}.$$

**Proof** We have for any episode $k$ and any transition $P' \in \mathcal{P}_k$,

$$\left|P' \left(x'|x,a\right) - P\left(x'|x,a\right)\right| \le \sqrt{\frac{2P' \left(x'|x,a\right) \left(1 - P'\left(x'|x,a\right)\right) \ln \iota}{\widehat{N}_k(x,a)}} + \frac{4E_{\varepsilon,\delta} + 7\ln \iota}{\widehat{N}_k(x,a)}$$

$$\le \sqrt{\frac{2\ln \iota}{\widehat{N}_k(x,a)}} \cdot \left(\sqrt{P\left(x'|x,a\right) \left(1 - P\left(x'|x,a\right)\right)} + 3.8\sqrt{\frac{\ln \iota}{\widehat{N}_k(x,a)}} + 1.5\sqrt{\frac{4E_{\varepsilon,\delta} + 7\ln \iota}{\widehat{N}_k(x,a)}}\right) + \frac{4E_{\varepsilon,\delta} + 7\ln \iota}{\widehat{N}_k(x,a)}$$

$$\le \sqrt{\frac{2P\left(x'|x,a\right) \left(1 - P\left(x'|x,a\right)\right) \ln \iota}{\widehat{N}_k(x,a)}} + \frac{7E_{\varepsilon,\delta} + 26\ln \iota}{\widehat{N}_k(x,a)},$$

where we apply Lemma E.8 to Lemma E.14 for the second step, and the third step follows from the fact that $\sqrt{x+y} \le \sqrt{x} + \sqrt{y}$ and $\sqrt{xy} \le \frac{1}{2}(x+y)$ for any $x, y \ge 0$. ∎

**Corollary E.16** *Conditioning on event in Lemma E.14, it holds for any episode $k$ and any transition $P' \in \mathcal{P}_k$,*

$$\left\| P'\left(\cdot | x, a\right) - P\left(\cdot | x, a\right) \right\|_1 \le \min\left(2, \sqrt{\frac{2C_{x,a} \ln \iota}{\widehat{N}_k(x,a)}} + \frac{X_{h(x)+1}\left(7E_{\varepsilon,\delta} + 26 \ln \iota\right)}{\widehat{N}_k(x,a)}\right),$$

*where for all $(x, a)$ as $C_{x,a} := \left(\sum_{x' \in \mathcal{X}_{h(x)+1}} \sqrt{P\left(x' | x, a\right)\left(1 - P\left(x' | x, a\right)\right)}\right)^2$ from Bourel et al. [2020].*

**Proof** We introduce *local effective support* as $C_{x,a}$ from Bourel et al. [2020]. We have for any episode $k$ and any transition $P' \in \mathcal{P}_k$,

$$\left\| P'\left(\cdot | x, a\right) - P\left(\cdot | x, a\right) \right\|_1 \le \sum_{x' \in \mathcal{X}_{h(x)+1}} \left(\sqrt{\frac{2P\left(x' | x, a\right)\left(1 - P\left(x' | x, a\right)\right) \ln \iota}{\widehat{N}_k(x,a)}} + \frac{7E_{\varepsilon,\delta} + 26 \ln \iota}{\widehat{N}_k(x,a)}\right)$$

$$\le \sqrt{\frac{2C_{x,a} \ln \iota}{\widehat{N}_k(x,a)}} + \frac{X_{h(x)+1}\left(7E_{\varepsilon,\delta} + 26 \ln \iota\right)}{\widehat{N}_k(x,a)}.$$

∎

**Lemma E.17 (Local effective support, Lemma 4 in Bourel et al. [2020])** *For any state-action pair $(x, a)$, the local effective support is bounded,*

$$C_{x,a} \le X_{h(x)+1} - 1.$$

**Lemma E.18 (Lower bound of upper occupancy measure)** *For any episode $k$ and state $x \ne x_H$, it always holds that $u_k(x) \ge \frac{1}{XK}$.*

**Proof** Similar to Lemma D.28 in Jin et al. [2023], one can construct a specific transition $\hat{P} \in \mathcal{P}_k$, such that $q^{\hat{P}, \pi}(x) \ge \frac{1}{XK}$ for any $\pi$ given episode $k$ and $x$, which suffices due to the definition of $u_k(x)$.

For any tuple $(x, a, x') \in W_h$ and $h = 0, \cdots, H-1$, $\hat{P}(x'|x,a) = \widetilde{P}_k(x'|x,a) \cdot \left(1 - \frac{1}{K}\right) + \frac{1}{X_{h+1}K}$. It is easy to verify that $\hat{P}$ is a valid transition function and belongs to the confidence set $\mathcal{P}_k$. Finally, we have the lower bound of the upper occupancy measure,

$$q^{\hat{P}, \pi}(x) = \sum_{u \in \mathcal{X}_{h-1}} \sum_{v \in \mathcal{A}} q^{\hat{P}, \pi}(u, v) \hat{P}(x'|u, v) \ge \sum_{u \in \mathcal{X}_{h-1}} \sum_{v \in \mathcal{A}} q^{\hat{P}, \pi}(u, v) \cdot \frac{1}{XK} = \frac{1}{XK}.$$

∎

### E.3   DIFFERENCE LEMMA

**Lemma E.19 (Occupancy Measure Difference, Lemma D.3.1 of Jin et al. [2021])** *For any transition functions $P_1, P_2$ and any policy $\pi$,*

$$q^{P_1, \pi}(x) - q^{P_2, \pi}(x) = \sum_{h=0}^{h(x)-1} \sum_{(u,v,w) \in W_h} q^{P_1, \pi}(u, v)\left(P_1(w|u, v) - P_2(w|u, v)\right) q^{P_2, \pi}(x|w)$$

$$= \sum_{h=0}^{h(x)-1} \sum_{(u,v,w)\in W_h} q^{P_2,\pi}(u,v) \left(P_1(w|u,v) - P_2(w|u,v)\right) q^{P_1,\pi}(x|w),$$

where $q^{P',\pi}(x|w)$ *is the probability of visiting $x$ starting $w$ under policy $\pi$ and transition $P'$.*

**Lemma E.20 (Lemma C.5 of Dann et al. [2023])** *For any policies $\pi_1, \pi_2$ and any transition function $P, P$,*

$$\sum_{x\neq x_H} \sum_{a\in\mathcal{A}} \left| q^{P,\pi_1}(x,a) - q^{P,\pi_2}(x,a) \right| \leq H \sum_{x\neq x_H} \sum_{a\in\mathcal{A}} q^{P,\pi_1}(x) \left| \pi_1(a|x) - \pi_2(a|x) \right|,$$

Following the same idea in the proofs of Lemma C.4 in Dann et al. [2023], and Lemma D.3.8 in Jin et al. [2023], we consider a tight bound of the difference between occupancy measures in the following lemma.

**Lemma E.21** *Suppose the event in Lemma 3.1 holds. For any state $x \neq x_H$, episode $k$ and transition $P' \in \mathcal{P}_k$, policy $\pi_k$, we have*

$$\left| q^{P',\pi_k}(x) - q^{P,\pi_k}(x) \right| \leq \mathcal{O}\left( \sum_{h=0}^{h(x)-1} \sum_{(u,v,w)\in W_h} q^{P,\pi_k}(u,v) \cdot \sqrt{\frac{P(w|u,v)(1-P(w|u,v))\ln\iota}{\widehat{N}_k(u,v)}} \cdot q^{P,\pi_k}(x|w) \right)$$

$$+ \mathcal{O}\left( X^2 \sum_{u\neq x_H} \sum_{v\in\mathcal{A}} q^{P,\pi_k}(u,v) \cdot \frac{E_{\varepsilon,\delta} + \ln\iota}{\widehat{N}_k(u,v)} \right).$$

**Proof** According to Lemma E.19, we have

$$\left| q^{P',\pi_k}(x) - q^{P,\pi_k}(x) \right| \leq \sum_{h=0}^{h(x)-1} \sum_{(u,v,w)\in W_h} q^{P,\pi_k}(u,v) \left| P'(w|u,v) - P(w|u,v) \right| q^{P',\pi_k}(x|w)$$

$$= \sum_{h=0}^{h(x)-1} \sum_{(u,v,w)\in W_h} q^{P,\pi_k}(u,v) \left| P'(w|u,v) - P(w|u,v) \right| q^{P,\pi_k}(x|w)$$

$$+ \sum_{h=0}^{h(x)-1} \sum_{(u,v,w)\in W_h} q^{P,\pi_k}(u,v) \left| P'(w|u,v) - P(w|u,v) \right| \left( q^{P',\pi_k}(x|w) - q^{P,\pi_k}(x|w) \right)$$

$$\leq \underbrace{\sum_{h=0}^{h(x)-1} \sum_{(u,v,w)\in W_h} q^{P,\pi_k}(u,v) \left| P'(w|u,v) - P(w|u,v) \right| q^{P,\pi_k}(x|w)}_{\text{TERM (A)}}$$

$$+ \underbrace{\sum_{h=0}^{h(x)-1} \sum_{(u,v,w)\in W_h} q^{P,\pi_k}(u,v) \left| P'(w|u,v) - P(w|u,v) \right| \sum_{h'=h+1}^{h(x)-1} \sum_{(m,n,o)\in W_{h'}} q^{P,\pi_k}(m,n|w) \left| P'(o|m,n) - P(o|m,n) \right|}_{\text{TERM (B)}}$$

$$\leq \mathcal{O}\left( \sum_{h=0}^{h(x)-1} \sum_{(u,v,w)\in W_h} q^{P,\pi_k}(u,v) \cdot \sqrt{\frac{P(w|u,v)(1-P(w|u,v))\ln\iota}{\widehat{N}_k(u,v)}} \cdot q^{P,\pi_k}(x|w) \right) \qquad \text{(TERM (A.1))}$$

$$+ \mathcal{O}\left( \sum_{h=0}^{h(x)-1} \sum_{(u,v,w)\in W_h} q^{P,\pi_k}(u,v) \cdot \frac{E_{\varepsilon,\delta} + \ln\iota}{\widehat{N}_k(u,v)} \cdot q^{P,\pi_k}(x|w) \right) \qquad \text{(TERM (A.2))}$$

$$+ \mathcal{O}\Bigg( \sum_{h=0}^{h(x)-1} \sum_{(u,v,w)\in W_h} q^{P,\pi_k}(u,v) \cdot \sqrt{\frac{P(w|u,v)(1-P(w|u,v))\ln\iota}{\widehat{N}_k(u,v)}}$$

$$\cdot \sum_{h'=h+1}^{h(x)-1} \sum_{(m,n,o)\in W_{h'}} q^{P,\pi_k}(m,n|w) \sqrt{\frac{P(o|m,n)(1-P(o|m,n))\ln\iota}{\widehat{N}_k(m,n)}} \Bigg) \qquad \text{(TERM (B.1))}$$

$$+\mathcal{O}\left(\sum_{h=0}^{h(x)-1}\sum_{(u,v,w)\in W_h}q^{P,\pi_k}(u,v)\cdot\sqrt{\frac{P(w|u,v)(1-P(w|u,v))\ln\iota}{\widehat{N}_k(u,v)}}\right.$$

$$\left.\cdot\sum_{h'=h+1}^{h(x)-1}\sum_{(m,n,o)\in W_{h'}}q^{P,\pi_k}(m,n|w)\cdot\frac{E_{\varepsilon,\delta}+\ln\iota}{\widehat{N}_k(m,n)}\right) \qquad \text{(TERM (B.2))}$$

$$+\mathcal{O}\left(\sum_{h=0}^{h(x)-1}\sum_{(u,v,w)\in W_h}q^{P,\pi_k}(u,v)\cdot\frac{E_{\varepsilon,\delta}+\ln\iota}{\widehat{N}_k(u,v)}\cdot\sum_{h'=h+1}^{h(x)-1}\sum_{(m,n,o)\in W_{h'}}q^{P,\pi_k}(m,n|w)\right), \qquad \text{(TERM (B.3))}$$

where the second step firstly subtracts and adds $q^{P,\pi_k}(x|w)$ and then applies Lemma E.19 again for $\left|q^{P',\pi_k}(x|w)-q^{P,\pi_k}(x|w)\right|$ to obtain TERM (A) and TERM (B). Following Lemma E.15, we can decompose them into five terms as TERM (A.1), (A.2), and TERM (B.1), (B.2), (B.3). Then, we bound these terms separately.

Clearly, we can bound TERM (A.2) by letting $x$ be $x_H$,

$$\sum_{h=0}^{h(x)-1}\sum_{(u,v,w)\in W_h}q^{P,\pi_k}(u,v)\cdot\frac{E_{\varepsilon,\delta}+\ln\iota}{\widehat{N}_k(u,v)}\cdot q^{P,\pi_k}(x|w)\le X\sum_{x\ne x_H}\sum_{a\in\mathcal{A}}q^{P,\pi_k}(u,v)\cdot\frac{E_{\varepsilon,\delta}+\ln\iota}{\widehat{N}_k(u,v)}.$$

For TERM (B.1), we have

$$\text{TERM (B.1)}=\sum_{h=0}^{h(x)-1}\sum_{(u,v,w)\in W_h}\sum_{h'=h+1}^{h(x)-1}\sum_{(m,n,o)\in W_{h'}}q^{P,\pi_k}(u,v)\cdot\sqrt{\frac{P(w|u,v)(1-P(w|u,v))\ln\iota}{\widehat{N}_k(u,v)}}$$

$$\cdot q^{P,\pi_k}(m,n|w)\sqrt{\frac{P(o|m,n)(1-P(o|m,n))\ln\iota}{\widehat{N}_k(m,n)}}$$

$$\le\sum_{h=0}^{h(x)-1}\sum_{(u,v,w)\in W_h}\sum_{h'=h+1}^{h(x)-1}\sum_{(m,n,o)\in W_{h'}}q^{P,\pi_k}(u,v)\cdot\frac{P(w|u,v)(1-P(o|m,n))\ln\iota}{\widehat{N}_k(m,n)}\cdot q^{P,\pi_k}(m,n|w)$$

$$+\sum_{h=0}^{h(x)-1}\sum_{(u,v,w)\in W_h}\sum_{h'=h+1}^{h(x)-1}\sum_{(m,n,o)\in W_{h'}}q^{P,\pi_k}(u,v)\cdot q^{P,\pi_k}(m,n|w)\cdot\frac{P(o|m,n)(1-P(w|u,v))\ln\iota}{\widehat{N}_k(u,v)}$$

$$=\sum_{h=0}^{h(x)-1}\sum_{(m,n,o)\in W_h}\frac{(1-P(o|m,n))\ln\iota}{\widehat{N}_k(m,n)}\left(\sum_{h'=0}^{h(x)-1}\sum_{(u,v,w)\in W_{h'}}q^{P,\pi_k}(u,v)\cdot P(w|u,v)\cdot q^{P,\pi_k}(m,n|w)\right)$$

$$+\sum_{h=0}^{h(x)-1}\sum_{(u,v,w)\in W_h}\frac{q^{P,\pi_k}(u,v)(1-P(w|u,v))\ln\iota}{\widehat{N}_k(u,v)}\left(\sum_{h'=h+1}^{h(x)-1}\sum_{(m,n,o)\in W_{h'}}q^{P,\pi_k}(m,n|w)\cdot P(o|m,n)\right)$$

$$\le\sum_{h=0}^{h(x)-1}\sum_{(m,n,o)\in W_h}\frac{Hq^{P,\pi_k}(m,n)\ln\iota}{\widehat{N}_k(m,n)}+\sum_{h=0}^{h(x)-1}\sum_{(u,v,w)\in W_h}\frac{Hq^{P,\pi_k}(u,v)\ln\iota}{\widehat{N}_k(u,v)}$$

$$\le\mathcal{O}\left(HX\sum_{h=0}^{H-1}\sum_{x\in\mathcal{X}_h}\sum_{a\in\mathcal{A}}\frac{q^{P,\pi_k}(x,a)\ln\iota}{\widehat{N}_k(x,a)}\right),$$

where the second step applies $\sqrt{xy}\le x+y$ for any $x,y\ge0$; the third step rearranges the summation order, and the fourth step follows the facts that $\sum_{(u,v,w)\in W_{h'}}q^{P,\pi_k}(u,v)\cdot P(w|u,v)\cdot q^{P,\pi_k}(m,n|w)=q^{P,\pi_k}(m,n)$ and $\sum_{m\in\mathcal{X}_h'}\sum_{a\in\mathcal{A}}q^{P,\pi_k}(m,n|w)\cdot P(o|m,n)=q^{P,\pi_k}(o|w)$.

Similarly, we have TERM (B.2) bounded as

$$\text{TERM (B.2)} = \sum_{h=0}^{h(x)-1} \sum_{(u,v,w)\in W_h} \sum_{h'=h+1}^{h(x)-1} \sum_{(m,n,o)\in W_{h'}} q^{P,\pi_k}(u,v) \cdot \sqrt{\frac{P(w|u,v)(1-P(w|u,v))\ln\iota}{\widehat{N}_k(u,v)}}$$

$$\cdot q^{P,\pi_k}(m,n|w) \cdot \frac{E_{\varepsilon,\delta}+\ln\iota}{\widehat{N}_k(m,n)}$$

$$\leq \sum_{h=0}^{h(x)-1} \sum_{(u,v,w)\in W_h} \sum_{h'=h+1}^{h(x)-1} \sum_{(m,n,o)\in W_{h'}} q^{P,\pi_k}(u,v)\cdot P(w|u,v)\cdot q^{P,\pi_k}(m,n|w) \cdot \frac{E_{\varepsilon,\delta}+\ln\iota}{\widehat{N}_k(m,n)}$$

$$+ \sum_{h=0}^{h(x)-1} \sum_{(u,v,w)\in W_h} \sum_{h'=h+1}^{h(x)-1} \sum_{(m,n,o)\in W_{h'}} q^{P,\pi_k}(u,v)\cdot \frac{(1-P(w|u,v))\ln\iota}{\widehat{N}_k(u,v)} \cdot q^{P,\pi_k}(m,n|w)$$

$$\leq \sum_{h'=0}^{H-1} \sum_{(m,n,o)\in W_{h'}} \frac{E_{\varepsilon,\delta}+\ln\iota}{\widehat{N}_k(m,n)} \left(\sum_{h=0}^{h(x)-1} \sum_{(u,v,w)\in W_h} q^{P,\pi_k}(u,v)\cdot P(w|u,v)\cdot q^{P,\pi_k}(m,n|w)\right)$$

$$+ \sum_{h=0}^{h(x)-1} \sum_{(u,v,w)\in W_h} \frac{q^{P,\pi_k}(u,v)(1-P(w|u,v))\ln\iota}{\widehat{N}_k(u,v)} \left(\sum_{h'=h+1}^{h(x)-1} \sum_{(m,n,o)\in W_{h'}} q^{P,\pi_k}(m,n|w)\right)$$

$$\leq \sum_{h'=0}^{H-1} \sum_{(m,n,o)\in W_{h'}} \sum_{h=0}^{h(x)-1} q^{P,\pi_k}(m,n)\cdot \frac{E_{\varepsilon,\delta}+\ln\iota}{\widehat{N}_k(m,n)} + X \sum_{h=0}^{h(x)-1} \sum_{(u,v,w)\in W_h} \frac{q^{P,\pi_k}(u,v)\ln\iota}{\widehat{N}_k(u,v)}$$

$$\leq HX \sum_{x\neq x_H} \sum_{a\in\mathcal{A}} q^{P,\pi_k}(x,a) \frac{E_{\varepsilon,\delta}+\ln\iota}{\widehat{N}_k(x,a)} + X^2 \sum_{x\neq x_H} \sum_{a\in\mathcal{A}} \frac{q^{P,\pi_k}(x,a)\ln\iota}{\widehat{N}_k(x,a)},$$

where the first inequality uses the fact that $\widehat{N}_k(x,a)_k(x,a) \geq \mathcal{O}(E_{\varepsilon,\delta}+\ln\iota)$ according to the definition in Eq.D.1; the second inequality follows the facts that $\sum_{(u,v,w)\in W_h} q^{P,\pi_k}(u,v)\cdot P(w|u,v)\cdot q^{P,\pi_k}(m,n|w) = q^{P,\pi_k}(m,n)$ and $\sum_{m\in\mathcal{X}_{h'}} \sum_{a\in\mathcal{A}} q^{P,\pi_k}(m,n|w) \leq 1$. For TERM (B.3), we have

$$\text{TERM (B.3)} = \sum_{h=0}^{h(x)-1} \sum_{(u,v,w)\in W_h} q^{P,\pi_k}(u,v)\cdot \frac{E_{\varepsilon,\delta}+\ln\iota}{\widehat{N}_k(u,v)} \cdot \sum_{h'=h+1}^{h(x)-1} \sum_{(m,n,o)\in W_{h'}} q^{P,\pi_k}(m,n|w)$$

$$= \sum_{h=0}^{h(x)-1} \sum_{(u,v,w)\in W_h} q^{P,\pi_k}(u,v)\cdot \frac{E_{\varepsilon,\delta}+\ln\iota}{\widehat{N}_k(u,v)} \cdot \left(\sum_{h'=h+1}^{h(x)-1} \sum_{o\in\mathcal{X}_{h'+1}} 1\right)$$

$$\leq X^2 \sum_{u\neq x_H} \sum_{v\in\mathcal{A}} q^{P,\pi_k}(u,v)\cdot \frac{E_{\varepsilon,\delta}+\ln\iota}{\widehat{N}_k(u,v)}.$$

Putting all the bounds for these terms together yields the bound of the lemma,

$$\left| q^{P',\pi_k}(x) - q^{P,\pi_k}(x)\right| \leq \mathcal{O}\left(\sum_{h=0}^{h(x)-1} \sum_{(u,v,w)\in W_h} q^{P,\pi_k}(u,v)\cdot \sqrt{\frac{P(w|u,v)(1-P(w|u,v))\ln\iota}{\widehat{N}_k(u,v)}}\cdot q^{P,\pi_k}(x|w)\right)$$

$$+ \mathcal{O}\left(X^2 \sum_{u\neq x_H} \sum_{v\in\mathcal{A}} q^{P,\pi_k}(u,v)\cdot \frac{E_{\varepsilon,\delta}+\ln\iota}{\widehat{N}_k(u,v)}\right).$$

$\blacksquare$

## E.4 ESTIMATION ERROR

**Lemma E.22 (Lemma 10 in Jin et al. [2020a])** *With probability at least $1 - \delta$, we have for all $h \in [H]$,*

$$\sum_{k=1}^{K} \sum_{(x,a) \in \mathcal{X}_h \times \mathcal{A}} \frac{q^{P,\pi_k}(x,a)}{\max\{N_k(x,a), 1\}} = \mathcal{O}\left(X_h A \log K + \log\left(\frac{H}{\delta}\right)\right),$$

*and*

$$\sum_{k=1}^{K} \sum_{(x,a) \in \mathcal{X}_h \times \mathcal{A}} \frac{q^{P,\pi_k}(x,a)}{\sqrt{\max\{N_k(x,a), 1\}}} = \mathcal{O}\left(\sqrt{X_h A K} + X_h A \log K + \log\left(\frac{H}{\delta}\right)\right).$$

**Proposition E.23** *Let $\mathcal{E}_{EST}$ be the event such that we have for all $h \in [H]$ simultaneously*

$$\sum_{k=1}^{K} \sum_{(x,a) \in \mathcal{X}_h \times \mathcal{A}} \frac{q^{P,\pi_k}(x,a)}{\max\{N_k(x,a), 1\}} = \mathcal{O}\left(X_h A \log K + \log \iota\right),$$

*and*

$$\sum_{k=1}^{K} \sum_{(x,a) \in \mathcal{X}_h \times \mathcal{A}} \frac{q^{P,\pi_k}(x,a)}{\sqrt{\max\{N_k(x,a), 1\}}} = \mathcal{O}\left(\sqrt{X_h A K} + X_h A \log K + \log \iota\right).$$

*We have $\mathbb{P}\left[\mathcal{E}_{EST}\right] \geq 1 - \delta$.*

**Proof** The proof directly follows from the definition of $\iota$, which ensures that $\iota \geq \frac{H}{\delta}$. ∎

Based on the event $\mathcal{E}_{EST}$, we introduce the following lemma which is critical in analyzing the estimation error.

**Lemma E.24** *Suppose the event $\mathcal{E}_{EST}$ defined in Proposition E.23 holds. Then we have for all $h \in [H]$,*

$$\sum_{k=1}^{K} \sum_{(x,a) \in \mathcal{X}_h \times \mathcal{A}} \frac{q^{P,\pi_k}(x,a)}{\widehat{N}_k(x,a)} = \mathcal{O}\left(X_h A \log K + \log \iota\right),$$

*and*

$$\sum_{k=1}^{K} \sum_{(x,a) \in \mathcal{X}_h \times \mathcal{A}} \frac{q^{P,\pi_k}(x,a)}{\sqrt{\widehat{N}_k(x,a)}} \sqrt{C_{x,a}} \leq \mathcal{O}\left(\sqrt{\sum_{(x,a) \in \mathcal{X}_h \times \mathcal{A}} C_{x,a} K} + \sqrt{X_{h+1}} X_h A \log K + \sqrt{X_{h+1}} \log \iota\right)$$

**Proof** Since $\widehat{N}_k(x,a) \geq N_k(x,a)$ always holds by definition, the first equation directly follows Proposition E.23.

For the second equation, similar to the proof of Lemma 10 in Jin et al. [2020a], we decompose the term as

$$\sum_{k=1}^{K} \sum_{(x,a) \in \mathcal{X}_h \times \mathcal{A}} \frac{q^{P,\pi_k}(x,a)}{\sqrt{\widehat{N}_k(x,a)}} \sqrt{C_{x,a}} \leq \sum_{k=1}^{K} \sum_{(x,a) \in \mathcal{X}_h \times \mathcal{A}} \frac{q^{P,\pi_k}(x,a)}{\sqrt{N_k(x,a)}} \sqrt{C_{x,a}}$$

$$= \sum_{k=1}^{K} \sum_{(x,a) \in \mathcal{X}_h \times \mathcal{A}} \frac{\mathbb{I}_k(x,a)}{\sqrt{N_k(x,a)}} \sqrt{C_{x,a}} + \sum_{k=1}^{K} \sum_{(x,a) \in \mathcal{X}_h \times \mathcal{A}} \frac{q^{P,\pi_k}(x,a) - \mathbb{I}_k(x,a)}{\sqrt{N_k(x,a)}} \sqrt{C_{x,a}}.$$

The first term is bounded by

$$\sum_{k=1}^{K} \sum_{(x,a) \in \mathcal{X}_h \times \mathcal{A}} \frac{\mathbb{I}_k(x,a)}{\sqrt{N_k(x,a)}} \sqrt{C_{x,a}} = \mathcal{O}\left(\sum_{(x,a) \in \mathcal{X}_h \times \mathcal{A}} \sqrt{C_{x,a}} \cdot \sqrt{N_K(x,a)}\right)$$

$$\leq \mathcal{O}\left(\sqrt{\sum_{(x,a)\in\mathcal{X}_h\times\mathcal{A}} C_{x,a}K}\right), \tag{E.-6}$$

according to Lemma E.9 and Cauchy-Schwarz inequality.

For the second term, we apply Lemma E.10 with $Y_k = \sum_{(x,a)\in\mathcal{X}_h\times\mathcal{A}} \frac{q^{P,\pi_k}(x,a)-\mathbb{I}_k(x,a)}{\sqrt{N_k(x,a)}} \leq 1, \lambda = 1$, and the fact

$$\mathbb{E}_k\left[Y_k^2\right] \leq \mathbb{E}_k\left[\left(\sum_{(x,a)\in\mathcal{X}_h\times\mathcal{A}} \frac{\mathbb{I}_k(x,a)}{\sqrt{N_k(x,a)}}\right)^2\right] = \sum_{(x,a)\in\mathcal{X}_h\times\mathcal{A}} \frac{q^{P,\pi_k}(x,a)}{N_k(x,a)},$$

and combine the upper bound of $C_{x,a}$ in Lemma E.17 and Lemma E.22,

$$\sum_{k=1}^{K}\sum_{(x,a)\in\mathcal{X}_h\times\mathcal{A}} \frac{q^{P,\pi_k}(x,a)-\mathbb{I}_k(x,a)}{\sqrt{N_k(x,a)}}\sqrt{C_{x,a}} \leq \sqrt{X_{h+1}}\cdot\left(\sum_{k=1}^{K}\sum_{(x,a)\in\mathcal{X}_h\times\mathcal{A}} \frac{q^{P,\pi_k}(x,a)}{N_k(x,a)} + \ln\iota\right).$$

Combining both terms, we prove the result. ∎

# F    MISSING DETAILS FOR SECTION 5

In this section, we present the proof of privacy guarantees in Section 5. Recall that for visitation counters, $N_k(x,a)$ is the original count, $\ddot{N}_k(x,a)$ is the noisy count after step (1) of both Privatizers and $\widetilde{N}_k(x,a)$ is the final private counts. For losses, $L_k(x,a)$ is the original cumulative loss in full-information setting, and $\ddot{\ell}_k(x,a)$ is the non-private loss estimator in bandit-feedback setting, and $\widetilde{L}_k(x,a)$ and $\tilde{\ell}_k(x,a)$ are the private version of $L_k(x,a)$ and $\ddot{\ell}_k(x,a)$.

## F.1    A VARIANT OF THE BINARY MECHANISM

Firstly, we introduce a variant of the Binary Mechanism (Algorithm 4), which has also been introduced in Agarwal and Singh [2017]. This variant of Binary Mechanism deals with a continual observation $\{\sigma_k\}_{k\in[K]}$ with each $\sigma_k \in [0,1]$. In each step, the mechanism releases a number denoted as $\widetilde{\Sigma}_k$, which is the private version of the sum of the previous $k$ observed number, i.e., $\Sigma_k = \sum_{i=1}^{k}\sigma_i$. Initialized with privacy budget level $\varepsilon$, this mechanism follows the properties in Lemma F.1.

**Lemma F.1 (Guarantees of the Variant of Binary Mechanism)** *The following claims hold true:*
(1) **Privacy**: *The sequence* $\left\{\widetilde{\Sigma}_k\right\}_{k=1}^{K}$ *is $\varepsilon$-differentially private.*
(2) **Distribution**: *For all $k \in [K]$, the injected noise $\widetilde{\Sigma}_k - \Sigma_k = \sum_{i=1}^{\lceil\log K\rceil} b_i$, and each $b_i$ is i.i.d. sampled from* $Lap\left(\log K/\varepsilon\right)$.

**Proof** Our privacy guarantee is established through the Binary Mechanism with Laplace noise, accompanied by the post-processing property as outlined in Dwork et al. [2014]. We introduce additional noise beyond the standard Binary Mechanism, ensuring the injection of an exact $\lceil\log K\rceil$ number of noise samples, in line with our distribution property. ∎

## F.2    PRIVACY GUARANTEES

First, we demonstrate that our private loss for the bandit-feedback setting simultaneously satisfies LDP and DP in the streaming setting.

**Proof** [Proof of Lemma 5.6] First, we focus on a single episode $k$, and consider the private version $\tilde{\ell}_k(x,a)$ of the observed loss $\ell_k(x,a)\mathbb{I}_k(x,a)$. This private version $\tilde{\ell}_k(x,a)$ is obtained from $\ell_k(x,a)\mathbb{I}_k(x,a)$ via the Laplace mechanism with noise level $\frac{3H}{\varepsilon}$. Since the sensitivity of the input function is 1, the Laplace mechanism satisfies $\frac{\varepsilon}{3H}$-DP and $\frac{\varepsilon}{3H}$-LDP (refer

---

**Algorithm 4** A Variant of the Binary Mechanism

---

**Input:** Time upper bound $K$, privacy budget $\varepsilon > 0$, online stream $\sigma \in [0,1]^K$, binary tree $\mathcal{B}$

**Initialization:** $\varepsilon' = \varepsilon/\log K$, noisy sum $\widetilde{\Sigma}_k = 0$, initialize the binary tree $\mathcal{B}$ over $K$ leaves with all nodes

1: **for** $k = 1$ to $K$ **do**
2:     Express $k$ in binary form: $s_k$, where $k = \sum_{j=1}^{\lceil \log_2 K \rceil} s_k(j) \cdot 2^{\lceil \log_2 K \rceil - j}$. For example, if $s_k = 110$, then $s_k(1) = 1, s_k(2) = 1, s_k(3) = 0$
3:     Populate the $s_k$-th entry of $\mathcal{B}$: $\mathcal{B}_{s_k} \leftarrow \sigma_k$
4:     Perturb the $s_k$-th entry of $\mathcal{B}$: $\hat{\mathcal{B}}_{s_k} \leftarrow \mathcal{B}_{s_k} + b_{s_k}$, where $b_{s_k} \sim \text{Lap}\left(\frac{1}{\varepsilon'}\right)$
5:     Let $S_k$ be the set of all ancestors $s$ of $s_k$ in the tree $\mathcal{B}$, such that all the leaves in the sub-tree rooted at $s$ are already populated
6:     **for** all $s \in S_k$ **do**
7:        Update the value of the node with the values of its children: $\mathcal{B}_s \leftarrow \mathcal{B}_{s \circ 0} + \mathcal{B}_{s \circ 1}$
8:        Perturb the value of the node: $\hat{\mathcal{B}}_s \leftarrow \mathcal{B}_s + b_s$, where $b_s \sim \text{Lap}\left(\frac{1}{\varepsilon'}\right)$
9:     **end for**
10:    **for** $j = 1$ to $\lceil \log_2 K \rceil$ **do**
11:      **if** $s_k(j) = 1$ **then**
12:        Form binary string $s^q = s_k(1) \circ s_k(2) \circ \cdots \circ s_k(j-1) \circ 0$ of length $j$
13:        $\widetilde{\Sigma}_k \leftarrow \widetilde{\Sigma}_k + \hat{\mathcal{B}}_{s^q}$
14:      **else**
15:        $\widetilde{\Sigma}_k \leftarrow \widetilde{\Sigma}_k + b$, where $b \sim \text{Lap}\left(\frac{1}{\varepsilon'}\right)$
16:      **end if**
17:    **end for**
18:    **output:** $\widetilde{\Sigma}_k$
19: **end for**

---

to Dwork et al. [2014]). Additionally, since every episode involves at most $H$ bandit losses, according to [Hsu et al., 2014, Lemma 34], the composition of all these $XAK$ different Laplace mechanisms are $\frac{\varepsilon}{3}$-LDP and $\frac{\varepsilon}{3}$-DP. ∎

Now, let's proceed to establish the proof for the JDP guarantee.

**Proof** [Proof of Lemma 5.1] Under the full-information setting, the release of $\left\{\ddot{N}_k(x,a)\right\}_{(k,x,a)}$ satisfies $\frac{\varepsilon}{3}$-DP according to [Chan et al., 2011, Theorem 3.5] and [Hsu et al., 2014, Lemma 34]. Similarly, the releases of $\left\{\ddot{N}_k(x,a,x')\right\}_{(k,x,a,x')}$ and $\left\{\ddot{L}_k(x,a)\right\}_{(k,x,a)}$ also satisfy $\frac{\varepsilon}{3}$-DP. Therefore, the release of all these counts satisfies $\varepsilon$-DP. Due to post-processing [Dwork et al., 2014, Proposition 2.1], the release of all private counts $\left\{\widetilde{N}_k(x,a)\right\}_{(k,x,a)}, \left\{\widetilde{N}_k(x,a,x')\right\}_{(k,x,a,x')}, \left\{\widetilde{L}_k(x,a)\right\}_{(k,x,a)}$ also satisfy $\varepsilon$-DP.

Under the bandit setting, and with the help of Lemma 5.6 and post-processing property, the release of all private counts $\left\{\widetilde{N}_k(x,a)\right\}_{(k,x,a)}, \left\{\widetilde{N}_k(x,a,x')\right\}_{(k,x,a,x')}, \left\{\widetilde{\ell}_k(x,a)\right\}_{(k,x,a)}$ also satisfy $\varepsilon$-DP.

For utility analysis, we analyze the visitation counters and losses, separately.

**Private visitation counters.** By applying [Chan et al., 2011, Theorem 3.6] and setting $\varepsilon' = \frac{\varepsilon}{3H \log K}$ in Binary mechanism, and using a union bound, we can establish that with probability $1 - 2\delta$, for all $(k, x, a, x')$,

$$\left|\ddot{N}_k(x,a,x') - N_k(x,a,x')\right| \le O\left(\frac{3H}{\varepsilon}\log^{1.5} K \log\left(\frac{X^2 AK}{\delta}\right)\right), \quad \left|\ddot{N}_k(x,a) - N_k(x,a)\right| \le O\left(\frac{3H}{\varepsilon}\log^{1.5} K \log\left(\frac{XAK}{\delta}\right)\right).$$

Referring to the post-processing procedures and Lemma 5.1 in Qiao and Wang [2023a], the Central Privatizer satisfies Assumption 2.3 with $E_{\varepsilon,\delta} = O\left(\frac{H}{\varepsilon}\log^{1.5} K \log\left(\frac{XAK}{\delta}\right)\right)$, and $N_k(x,a) \le \widetilde{N}_k(x,a) \le N_k(x,a) + E_{\varepsilon,\delta}$. Furthermore, in accordance with the constraints of the optimization problem, we also observe that $\ddot{N}_k(x,a) = \sum_{x' \in \mathcal{X}_{h+1}} \ddot{N}_k(x,a,x')$, which implies that $\widetilde{N}_k(x,a) = \sum_{x' \in \mathcal{X}_{h+1}} \widetilde{N}_k(x,a,x')$.

**Private loss in full-information setting.** In the full-information setting, we have constructed a variant of the Binary Mechanism (Algorithm 4) to obtain $\widetilde{L}_k(x, a)$ in Section F.1. Since the cumulative loss $L_k(x, a)$ is privatized by this variant with privacy budget $\varepsilon/3H$, the injected noise $Z_k(x, a)$ is a sample of the summation of $\lceil \log K \rceil$ i.i.d. Laplace variables with parameter for all $(k, x, a)$. Therefore, for any episode $k$, when $K > \sqrt{XA}$, we have $\mathbb{E}\left[\max_{x,a} Z_k(x, a) - \min_{x,a} Z_k(x, a)\right] \leq \mathcal{O}\left(\frac{H}{\varepsilon}\sqrt{\log^3 K \ln(XA)}\right)$, which follows the maxima property of the sum of i.i.d. Laplace variables as demonstrated in Lemma E.7.

**Private loss in bandit-feedback setting** Under the bandit setting, we directly apply the Laplace Mechanism, and the noise injected is $Z_k(x, a) = \tilde{\ell}_k(x, a) - \ell_k(x, a)\mathbb{I}_k(x, a)$, where $Z_k(x, a) \sim \text{Lap}(3H/\varepsilon)$. Then using the concentration of the Laplace variable (refer to Lemma E.2) and a union bound, we can conclude that with probability at least $1 - \delta$, $\left|\tilde{\ell}_k(x, a) - \ell_k(x, a)\mathbb{I}_k(x, a)\right| \leq \frac{3H}{\varepsilon}\ln\left(\frac{XAK}{\delta}\right)$ for all $(k, x, a)$. ∎

**Proof** [Proof of Theorem 5.2] With $\varepsilon$-DP guarantee for private counters and losses, the release of all $\pi_k$ is also $\varepsilon$-DP according to post-processing property under both full-information and bandit settings. Finally, the guarantee of $\varepsilon$-JDP for final action sequences follows the Billboard Lemma [Hsu et al., 2014, Lemma 9]. Besides, the regret bound is obtained by plugging $E_{\varepsilon,\delta}$, and corresponding $E_\varepsilon, E'_{\varepsilon,\delta}$ in Lemma 5.1 into Theorem 3.3 and Theorem 4.2. ∎

**Proof** [Proof of Lemma 5.7] The privacy guarantee directly results from properties of Laplace Mechanism and composition of DP [Dwork et al., 2014], and Lemma 5.6. For utility analysis, we also analyze the visitation counters and losses as below.

**Private visitation counters.** According to Dwork et al. [2014] Corollary 12.4 and a union bound, with probability $1 - 2\delta$, for all $(k, x, a, x')$,

$$\left|\ddot{N}_k(x, a, x') - N_k(x, a, x')\right| \leq \mathcal{O}\left(\frac{3H}{\varepsilon}\sqrt{K\log\left(\frac{X^2AK}{\delta}\right)}\right), \quad \left|\ddot{N}_k(x, a) - N_k(x, a)\right| \leq \mathcal{O}\left(\frac{3H}{\varepsilon}\sqrt{K\log\left(\frac{XAK}{\delta}\right)}\right). \tag{F.1}$$

Together with Lemma 5.5, the Local Privatizer satisfies Assumption 2.3 with $E_{\varepsilon,\delta} = \mathcal{O}\left(\frac{H}{\varepsilon}\sqrt{K\log\left(\frac{XAK}{\delta}\right)}\right)$.

**Private loss in full-information setting.** Under the full-information setting, we apply the Laplace Mechanism with privacy budget $\varepsilon' = \frac{\varepsilon}{3H}$ directly on the point-wise loss $\ell_k(x, a)$ and make summation to obtain $\widetilde{L}_k(x, a)$. Thus, the noise injected $Z_k(x, a)$ is a realization of a summation of $K$ Laplace variables for all $(k, x, a)$. Therefore, for any episode $k$, when $K > \frac{\ln(XA)}{2}$, $\mathbb{E}\left[\max_{x,a} Z_k(x, a) - \min_{x,a} Z_k(x, a)\right] \leq \mathcal{O}\left(\frac{H}{\varepsilon}\sqrt{K\ln(XA)}\right)$, which follows Lemma E.7.

**Private loss in bandit-feedback setting.** Under the bandit setting, the utility is the same as the JDP setting. ∎

**Proof** [Proof of Theorem 5.8] The regret bound is obtained by plugging $E_{\varepsilon,\delta}$, and corresponding $E_\varepsilon, E'_{\varepsilon,\delta}$ in Lemma 5.7 into Theorem 3.3 and Theorem 4.2. ∎