# OpenReview forum: "Differentially Private No-regret Exploration in Adversarial Markov Decision Processes"
_auai.org/UAI/2024/Conference — UAI 2024 poster_

### Official Review · Reviewer_3jR4 · 2024-03-16

**Q2-1 Originality-Novelty:** 2
**Q2-2 Correctness-Technical Quality:** 3
**Q2-5 Clarity Of Writing:** 3

**Q1 Summary And Contributions:**

This paper focuses on studying adversarial Markov Decision Processes with differential privacy guarantees, which prevent the discovery of personally sensitive data from communication. The author proposes two algorithms for full-information feedback and bandit feedback, with theoretically performance guarantees that match the existing lower regret bound.

**Q2-3 Extent To Which Claims Are Supported By Evidence:**

3: Good: the main claims are supported by convincing evidence (in the form of adequate experimental evaluation, proofs, (pseudo-)code, references, assumptions).

**Q2-4 Reproducibility:**

3: Good: key resources (e.g. proofs, code, data) are available and key details (e.g. proofs, experimental setup) are sufficiently well-described for competent researchers to confidently reproduce the main results.

**Q3 Main Strengths:**

1. This work provides the first guarantee for privacy protection in adversarial MDP settings.

2. The author provides algorithms for both full-information feedback and bandit feedback, with theoretically performance guarantees that match the existing lower regret bound.

3. The paper is well written and easy to follow.

**Q4 Main Weakness:**

1. The proposed framework relies on solving a constraint optimization on the occupancy measure. This process is time-consuming and not practical in real-world applications.

2. The assumption of non-overlapping seems too restrictive. Though we can always replace the state $s$ with a state-stage pair $(s,h)$ to avoid this problem, it will increase the state space by a factor of $H$. In this situation, the regret will also have an extra dependency on $\sqrt{H}$ and be larger than the lower bound. It is necessary to provide more explanation when claiming near-optimal regret.

3. It seems strange to directly introduce importance sampling for bandit feedback (4.2) without any modification. Though this technique is widely used in adversarial bandit problems, it usually adds a small constant value $\gamma$ in the occupancy measure, for example, ($l(x,a)/(u(x,a)+\gamma)$), to avoid blowing up when $u(x,a)$ is close to zero. In fact, when the magnitude of the loss function $l$ is unbounded, the original Follow the Regularized Leader may fail to provide efficient guarantees. It seems necessary to provide more explanation about this problem.

4. This work seems to be only a combination of existing analyses on aderversarial MDP and privacy mechanism, which lacks of novelty.

**Q5 Detailed Comments To The Authors:**

1. For the problem-dependent regret, does the effective support close to the variance of the transition process? There also exist several previous works that studied the Variance-Dependent regret bound for both tabular and linear situations, which seems close to the problem-dependent regret.

[1] Sharp Variance-Dependent Bounds in Reinforcement Learning: Best of Both Worlds in Stochastic and Deterministic Environments

[2] Variance-dependent regret bounds for linear bandits and reinforcement learning: Adaptivity and computational efficiency

2. In the related work section, the author overlooks recently published works that also provide LDP guarantees for linear mixture MDPs.

[1] Locally differentially private reinforcement learning for linear mixture markov decision processes

3. The author mentions several potential future directions, including privatized policy-optimization-based algorithms and MDPs with function approximation. It is worth noting that some existing works have studied policy optimization-based algorithms with linear adversarial MDPs [1,2]. It would be interesting if the author could extend the proposed privacy mechanism to these works.

[1] Provably efficient exploration in policy optimization

[2] Near-optimal policy optimization algorithms for learning adversarial linear mixture mdps

I will be happy to raise my score if my concerns are addressed.

**Q9 Complying With Reviewing Instructions:**

Yes

---

> ### Author Rebuttal · Authors · 2024-04-09
>
> **Q1. ''The proposed framework relies on solving a constraint optimization ...''**\
> Although our algorithm involves an optimization problem, it admits a time efficient implementation. Specifically, the main computational issue only arises in the implementation of FTRL and OMD (i.e., solving Eq.(3.2) and Eq.(4.3)). They can be addressed by solving an unconstrained optimization problem which has a closed-form solution, and then solving a projection problem, following the common two-step implementation procedure. The solution procedure of the projection problem is similar to the one in [2,5]. More specifically, both the projection problems in our paper and in [2,5] are convex optimization problems with linear constraints and thus can be solved in polynomial time.  Moreover, Slater’s condition in the primal problem holds, and therefore the strong duality holds. This allows one to solve the associated dual problem instead, which has only non-negativity constraints, and is thus easier to solve. (please see Appendix C for detailed discussions on the computational issue).
>
> **Q2. ''The assumption of non-overlapping seems too restrictive ... It is necessary to provide more explanation when claiming near-optimal regret.''**\
> This assumption (also known as layered, loop-free assumption) is a standard one in the adversarial MDP literature [1-5]. This assumption can be related to episodic RL setting in [6] (i.e., the one with stationary transitions and rewards across episode steps).  In fact, an episodic non-stationary MDP can be defined over a new state-space of size $X'=HX$ but satisfying the above loop-free and layered structure. We will include this discussion in the paper.
>
> **Q3. ''It seems strange to directly introduce importance sampling for bandit feedback (4.2) without any modification ...''.**\
> We believe the technique the reviewer mentioned is the _implicit exploration_ method [2,7], which was proposed to reduce the variance of the loss estimators and help achieve a high-probability regret bound (and also as a means to deal with an adaptive adversary). Note that we have assumed that the adversary chooses the loss functions _obliviously_, i.e., the loss functions are selected before the game starts and are independent with the policy $\pi_k$. When the policies $\pi_k$ are chosen randomly, the corresponding performance metric is the expected regret, and then our optimistic estimator $\hat{\ell}$ could work with upper confidence occupancy as importance weight. Besides, we also use OMD with a log-barrier regularizer instead of the classic negative entropy regularizer to perform policy updates, which has the benefit of keeping our algorithm more stable.
>
> **Q4. ''This work seems to be only a combination of existing analyses on adversarial MDP and privacy mechanism, which lacks novelty.''**\
> We kindly refer the reviewer to our response to Q2 of the reviewer YntM.
>
> **Q5. ''For the problem-dependent regret, does the effective support close to the variance of the transition process? ... ''**\
> Based on our reading, the papers mentioned by the reviewer provide problem-dependent bounds that regard the variance of the optimal value function. Different from them, our regret bounds adapt to the effective support, which describes the difficulty of underlying MDPs transition dynamics.  In some sense, the variance of the transition process is smaller, and the probability of reaching any state is closer, then this term will be larger. Besides, the local effective support of any $x,a$ is always bounded by the number of successors of $x,a$.
>
> Thanks for the support for our work and the suggestion on adding related work on the private linear mixture MDP and problem-dependent regrets. We will do so.
>
> 1. Neu et al. The adversarial stochastic shortest path problem with unknown transition probabilities. AISTATS 2012.
> 2. Jin et al. Learning adversarial Markov decision processes with bandit feedback and unknown transition. ICML, 2020.
> 3. Zhao et al. Learning adversarial linear mixture markov decision processes with bandit feedback and unknown transition. ICLR, 2023.
> 4. Liu et al. Towards Optimal Regret in Adversarial Linear MDPs with Bandit Feedback. arXiv, 2023.
> 5. Rosenberg et al. Online convex optimization in adversarial markov decision processes. ICML, 2019.
> 6. Azar et al. Minimax regret bounds for reinforcement learning. ICML, 2017.
> 7. Neu et al. Explore no more: Improved high-probability regret bounds for non-stochastic bandits. NeurIPS, 2015.

---

### Official Review · Reviewer_YntM · 2024-03-23

**Q2-1 Originality-Novelty:** 2
**Q2-2 Correctness-Technical Quality:** 3
**Q2-5 Clarity Of Writing:** 3

**Q1 Summary And Contributions:**

The paper studies adversarial MDP with privacy constraints. Under both full information and bandit feedback, the paper proposes a framework to efficiently minimize regret while preserving either joint DP or local DP.

**Q2-3 Extent To Which Claims Are Supported By Evidence:**

3: Good: the main claims are supported by convincing evidence (in the form of adequate experimental evaluation, proofs, (pseudo-)code, references, assumptions).

**Q2-4 Reproducibility:**

3: Good: key resources (e.g. proofs, code, data) are available and key details (e.g. proofs, experimental setup) are sufficiently well-described for competent researchers to confidently reproduce the main results.

**Q3 Main Strengths:**

The paper is clearly written. To my best knowledge, the paper proposed the first private algorithm in adversarial MDP, which attains sublinear regret under suitable choice of privacy parameter.

**Q4 Main Weakness:**

It's not very clear to me what is the most important application of private algorithms in adversarial MDP. Moreover, given that the proposed algorithm is based on OMD and there is already a lot of work on private OMD for JDP/LDP, I am not quite sure what's the main technical contribution of this work.  There's also no experiment to validate the experiment.

**Q5 Detailed Comments To The Authors:**

1. Could you elaborate on some real applications for private adversarial MDP algorithms?
2. Could you highlight the main technical contribution of the proposed algorithm?

**Q9 Complying With Reviewing Instructions:**

Yes

---

> ### Author Rebuttal · Authors · 2024-04-09
>
> Thank you for your valuable comments and suggestions. We provide our response to each question below.
>
> **Q1. Real applications for private adversarial MDP algorithms?**\
> Our work lies in the intersection between private (classical) online learning and private reinforcement learning, where each episode corresponds to a user. In other words, we learn from users' private data and feedback, and the feedback may vary across different users. The assumption of adversarial loss (or reward) functions is supported by two observations: (i) the loss function could vary across episodes in an unpredictable manner; (ii) a stochastic loss function might depend on extra variables that are controlled by some complex part of the environment that is hard to model and predict. Such extra variables could only influence the loss incurred by the user. Under both (i) and (ii), which can arise in many practical scenarios (with privacy concerns), modeling loss functions as adversarial would be more relevant; examples include inventory management discussed in [1], recommendation systems, and clinical trials. E.g., in recommendation systems [2], the agent recommends items (corresponding to actions) according to users' search input (corresponding to states), and improves its performance based on users' rating (corresponding to the non-stationary rewards), and the rating may reflect different preferences. In clinical trials [3], the doctor applies treatments (corresponding to actions) according to patients' physical condition (corresponding to states), and tries to cure more patients based on the the effect of treatments on different patients (corresponding to the rewards). The reward could depend on some complex and hard-to-model historical variable of each patient (other than physical condition). We include these examples as motivating applications in the paper.
>
> **Q2. Highlight the main technical contributions.**\
> **(a) Tighter confidence sets.** Compared with previous works studying adversarial MDPs, which only focus on the worst cases and ignore the hardness of the underlying MDPs, we introduce a tighter confidence bound on components of the transition function and hence occupancy measures. This allows us to derive refined regret bounds, which makes appear problem-dependent properties of the transition function in the bound under both full-information and bandit settings. This makes the bounds more informative and allows them to adapt to the difficulty of problem (in terms of involved transition dynamics). \
> **(b) Novel private and optimistic loss estimator for adversarial MDPs under the bandit setting.**  Compared with the works studying adversarial MDPs under the bandit setting whose loss estimator cannot be directly applied in our private setting, the private loss without any modification is unbounded and probably negative, which will make previous analysis fail. Compared with the work studying private adversarial bandits, they only focus on pure DP constraints and bandits problem. We generalize the private loss to adversarial MDP with Assumption 4.1, scale the private loss to $[0,1]$, and construct an optimistic loss estimator using an upper confidence occupancy measure. The private loss estimator can be deployed in both JDP and LDP constraints satisfying Lemma 5.1 and Lemma 5.7, which is non-trivial and could be of interest beyond this work. Besides, since traditional private FTRL with negative entropy regularizer will fail in REG analysis for RL due to an unbounded ''stability'' term, we introduce log-barrier regularizer for private OMD, which results in a bounded ''stability'' term and help the regret guarantee. \
> **(c) Novel Privatizers for adversarial RL (under assumptions).**  Compared with the works studying private RL with stochastic MDPs, our algorithms are similar to those works by maintaining a private estimation of transition function with confidence bound (note that our confidence bound is tighter).  One key difference is that we privatize adversarial loss functions, while previous works only focus on stationary loss functions.To our best knowledge, we are the first to maintain such non-trivial Privatizers for the loss functions of RL with adversarial MDPs under both full-information and bandit settings, which is non-trivial and could be of interest beyond this work. Deploying the Privatizers with desired properties in our algorithms ''Private UC-O-REPS'' (resp. ''Private UOB-LBPS'') for full-information (resp. bandit) setting lets us achieve no-regret with new analysis line.
>
> 1. Neu et al. The adversarial stochastic shortest path problem with unknown transition probabilities. AISTATS 2012.
> 2. Zhou et al. A privacy-preserving distributed contextual federated online learning framework with big data support in social recommender systems. TKDE, 2019.
> 3. Liu et al. Blockchain-enabled contextual online learning under local differential privacy for coronary heart disease diagnosis in mobile edge computing. JBHI, 2020.

---

### Official Review · Reviewer_eNpR · 2024-03-23

**Q2-1 Originality-Novelty:** 2
**Q2-2 Correctness-Technical Quality:** 3
**Q2-5 Clarity Of Writing:** 3

**Q1 Summary And Contributions:**

This paper studies differential private adversarial  MDPs with differential privacy (DP). They consider both the local DP model and global DP model. For the bandit feedback model, the (non-private) regret is near-optimal up to an extra $\sqrt{X}$ factor, where $X$ is the state size.

**Q2-3 Extent To Which Claims Are Supported By Evidence:**

4: Excellent: all claims are supported by very convincing evidence (in the form of comprehensive experimental evaluation, rigorous mathematical proofs, detailed (pseudo-)code, precise references, well-motivated and realistic assumptions) and the authors deliver what they promise.

**Q2-4 Reproducibility:**

3: Good: key resources (e.g. proofs, code, data) are available and key details (e.g. proofs, experimental setup) are sufficiently well-described for competent researchers to confidently reproduce the main results.

**Q3 Main Strengths:**

Theoretical analysis is non-trivial.

**Q4 Main Weakness:**

1. There is no private lower bounds. So, I cannot see how tight it is for the analysis.

2. Computationally expensive.

3. Since no discussions on the private regret lower bounds, we cannot see whether we can improve the term involving $\epsilon$ or not.

4. Since the classical tabular MDP setting is  a special case of adversarial MDP setting,  it will be good to add some discussions for the case where rewards are iid drawn.

**Q5 Detailed Comments To The Authors:**

n/a

**Q9 Complying With Reviewing Instructions:**

Yes

---

> ### Author Rebuttal · Authors · 2024-04-09
>
> Thank you for your valuable comments and suggestions. We provide our response to each question below.
>
> **Q1, Q3. ''There are no private lower bounds, ..., whether we can improve the term involving $\epsilon$ or not.''.**\
> Of existing lower bounds, the most relevant to our case are those for private stochastic RL presented in [1,2], which could be used for the bandit setting.
> Specifically, [2] reports a regret lower bound of  $\Omega(H\sqrt{XAK} + \frac{XAH\log K}{\epsilon})$ for episodic RL (stochastic rewards) under JDP, and [1] proves one scaling as $\Omega(\frac{H\sqrt{XAK}}{\min(e^\epsilon-1,1)})$ for a similar problem setting but under LDP.
>
> Under both lower bounds, the regret due to the non-private part scales as $\Omega(H\sqrt{XAK})$, whereas the corresponding non-private terms in our bounds (which are problem-dependent) are sub-optimal by a factor of $\sqrt{X}$ in the worst-case. Albeit sub-optimal, this term improves over the best existing results in the non-private adversarial MDP [3,4] by leveraging problem-dependence and also matches them in the worst case.
>
> For the privacy-related term, in the case of JDP, our bound has an optimal dependency on $\epsilon$ in view of the JDP lower bound (from [1]), but in terms of $K$, our bound is sub-optimal. This gap could be attributed to inefficiency of our algorithm but might also arise due to a loose lower bound. In the case of LDP, [2] implies a lower bound of $\Omega(\frac{H\sqrt{XAK}}{\epsilon})$, for the privacy-related term assuming small enough $\epsilon$ (corresponding to high privacy regime). Here, our bound has an optimal dependency on $\epsilon$ and $K$, but has a sub-optimality gap of $H\sqrt{X^7A}$, at most, which could partly be due to a loose lower bound.
>
> We will include this discussion in the paper.
>
> **Q2. ''Computationally Expensive.''**\
> Although our algorithm involves an optimization problem, it admits a time efficient implementation. Specifically, the main computational issue only arises in the implementation of FTRL and OMD (i.e., solving Eq.(3.2) and Eq.(4.3)). They can be addressed by solving an unconstrained optimization problem which has a closed-form solution, and then solving a projection problem, following the common two-step implementation procedure (see, e.g., Chapter 28 in [5]). The solution procedure of the projection problem is similar to the one in [3,4,6]. More specifically, both the projection problems in our paper and in [3,4,6] are convex optimization problems with linear constraints and thus can be solved in polynomial time. Moreover, Slater’s condition in the primal problem holds, and therefore the strong duality holds. This allows one to solve the associated dual problem instead, which has only non-negativity constraints, and is thus easier to solve. (please see Appendix C for detailed discussions on the computational issue).
>
> **Q4. ''Since the classical tabular MDP setting is a special case of adversarial MDP setting, it will be good to add some discussions for the case where rewards are iid drawn. ''**\
> If the reward functions happen to be stochastic (corresponding to iid rewards), our regret bounds remain valid but the algorithm may be unnecessarily conservative due to being designed for adversarial scenarios. We believe the right approach in the face of such situations -- i.e., when one seeks adaptivity to both iid and adversarial settings simultaneously -- is to use best-of-both-worlds algorithms (e.g., [7,8]) which require carefully designed regularizers and a particular analysis on proving a kind of self-bounding regret bounds. To our best knowledge, no private best-of-both-world algorithm exists in the literature (even for multi-armed bandits), but their design would be a promising future direction. We add this discussion and provide pointers to this future work direction.
>
> 1. Vietri et al. Private reinforcement learning with pac and regret guarantees. ICML, 2020.
> 2. Garcelon et al. Local differential privacy for regret minimization in reinforcement learning. NeurIPS, 2021.
> 3. Rosenberg et al. Online convex optimization in adversarial markov decision processes. ICML, 2019.
> 4. Jin et al. Learning adversarial Markov decision processes with bandit feedback and unknown transition. ICML, 2020.
> 5. Lattimore et al. Bandit algorithms. Cambridge University Press, 2020.
> 6. Lee et al. Bias no more: high-probability data-dependent regret bounds for adversarial bandits and mdps. NeurIPS 2020.
> 7. Jin et al. Simultaneously learning stochastic and adversarial episodic mdps with known transition. NeurIPS, 2020.
> 8. Zimmert et al. An optimal algorithm for stochastic and adversarial bandits. AISTATS, 2019.

---

### Official Review · Reviewer_LG3y · 2024-03-23

**Q2-1 Originality-Novelty:** 3
**Q2-2 Correctness-Technical Quality:** 3
**Q2-5 Clarity Of Writing:** 3

**Q1 Summary And Contributions:**

The paper considers Differential Privacy (DP) in adversarial MDPs and design mechanisms to privatize both the transition matrix and the loss function. They obtain regret gurantees by satisfying either of the two rquirements - Joint Dp where a central user protects privacy and Local DP where each user protects its own privacy.

**Q2-3 Extent To Which Claims Are Supported By Evidence:**

3: Good: the main claims are supported by convincing evidence (in the form of adequate experimental evaluation, proofs, (pseudo-)code, references, assumptions).

**Q2-4 Reproducibility:**

3: Good: key resources (e.g. proofs, code, data) are available and key details (e.g. proofs, experimental setup) are sufficiently well-described for competent researchers to confidently reproduce the main results.

**Q3 Main Strengths:**

- The paper is well written - provides adequate motivation and highlights the technical challenges involved.
- To the best of my knowledge, the proof sketches in the main paper seem fine.

**Q4 Main Weakness:**

- No major weaknesses

**Q5 Detailed Comments To The Authors:**

1. Could the authors explain/ put in perspective their regret bounds with regards to the lower bound in https://arxiv.org/abs/2010.11425 ?

**Q9 Complying With Reviewing Instructions:**

Yes

---

> ### Author Rebuttal · Authors · 2024-04-09
>
> Thank you for your valuable comments and suggestions. We provide our response to each question below.
>
> **Q. ''Could the authors explain in perspective their regret bounds with regards to the lower bound in ... ?''**\
> Based on our reading, the mentioned paper on private linear bandits does not provide any lower bound, but gives pointers to [1] which provides a regret lower bound on private linear bandits. More relevant to our case are the regret lower bounds for private stochastic RL presented in [2,3], which could be used for the bandit setting.
> Specifically, [2] reports a regret lower bound of  $\Omega(H\sqrt{XAK} + \frac{XAH\log K}{\epsilon})$ for episodic RL (stochastic rewards) under JDP, and [3] proves one scaling as $\Omega(\frac{H\sqrt{XAK}}{\min(e^\epsilon-1,1)})$ for a similar problem setting but under LDP.
>
> Under both lower bounds, the regret due to the non-private part scales as $\Omega(H\sqrt{XAK})$, whereas the corresponding non-private terms in our bounds (which are problem-dependent) are sub-optimal by a factor of $\sqrt{X}$ in the worst-case. Albeit sub-optimal, this term improves over the best existing results in the non-private adversarial MDP [4,5] by leveraging problem-dependence and also matches them in the worst case.
>
> For the privacy-related term, in the case of JDP, our bound has an optimal dependency on $\epsilon$ in view of the JDP lower bound (from [2]), but in terms of $K$, our bound is sub-optimal. This gap could be attributed to inefficiency of our algorithm but might also arise due to a loose lower bound. In the case of LDP, [3] implies a lower bound of $\Omega(\frac{H\sqrt{XAK}}{\epsilon})$, for the privacy-related term assuming small enough $\epsilon$ (corresponding to high privacy regime). Here, our bound has an optimal dependency on $\epsilon$ and $K$, but has a sub-optimality gap of $H\sqrt{X^7A}$, at most, which could partly be due to a loose lower bound.
>
> We will include this discussion in the paper.
>
> 1. Shariff et al. Differentially private contextual linear bandits. NeurIPS, 2018.
> 2. Vietri et al. Private reinforcement learning with pac and regret guarantees. ICML, 2020.
> 3. Garcelon et al. Local differential privacy for regret minimization in reinforcement learning. NeurIPS, 2021.
> 4. Rosenberg et al. Online convex optimization in adversarial markov decision processes. ICML, 2019.
> 5. Jin et al. Learning adversarial Markov decision processes with bandit feedback and unknown transition. ICML, 2020.

---

### Meta-Review · Area_Chair_Jaxg · 2024-04-25

Reviewers came to a consensus on this paper. It is generally clear and well written. Overall, I found the comments in the reviews to be slightly more positive than the average score would suggest.

The paper develops DP private algorithms for learning adversarial MDP. This is a new problem setting which requires some new ideas. The relationship between this work and prior work in related settings is clearly described and the authors' discussion responses provide further context that can improve a final version.

While some reviewers had some concerns about motivation, each of the individual components of the problem setting have previously received significant attention, so I do not find this to be a concern.